# Geometric and Physical Constraints Synergistically Enhance Neural PDE Surrogates

**Yunfei Huang** [1] [2]   **David S. Greenberg** [1] [2]

## Abstract

Neural PDE surrogates can improve the cost-accuracy tradeoff of classical solvers, but often generalize poorly to new initial conditions and accumulate errors over time. Physical and symmetry constraints have shown promise in closing this performance gap, but existing techniques for imposing these inductive biases are incompatible with the staggered grids commonly used in computational fluid dynamics. Here we introduce novel input and output layers that respect physical laws and symmetries on the staggered grids, and for the first time systematically investigate how these constraints, individually and in combination, affect the accuracy of PDE surrogates. We focus on two challenging problems: shallow water equations with closed boundaries and decaying incompressible turbulence. Compared to strong baselines, symmetries and physical constraints consistently improve performance across tasks, architectures, autoregressive prediction steps, accuracy measures, and network sizes. Symmetries are more effective than physical constraints, but surrogates with both performed best, even compared to baselines with data augmentation or pushforward training, while themselves benefiting from the pushforward trick. Doubly-constrained surrogates also generalize better to initial conditions and durations beyond the range of the training data, and more accurately predict real-world ocean currents.

[1]Helmholtz Centre Hereon, Geesthacht, Germany [2]Helmholtz AI. Correspondence to: Yunfei Huang <yunfei.huang5@gmail.com>, David S. Greenberg <david.greenberg@hereon.de>.

*Proceedings of the $42^{nd}$ International Conference on Machine Learning*, Vancouver, Canada. PMLR 267, 2025. Copyright 2025 by the author(s).

## 1. Introduction

Recently, neural networks have shown promising results in predicting the time evolution of PDE systems, often achieving cost-accuracy tradeoffs that outperform traditional numerical methods (Li et al., 2021; Gupta & Brandstetter, 2023; Stachenfeld et al., 2021; Takamoto et al., 2022; Long et al., 2019; Um et al., 2020; Kochkov et al., 2021). However, obtaining accurate and stable autoregressive 'rollouts' over long durations remains notoriously difficult. Several techniques have been proposed to address this problem, including physical constraints, symmetry equivariance, time-unrolled training, specialized architectures, data augmentation, addition of input noise and generative modeling (Sanchez-Gonzalez et al., 2020; Lippe et al., 2024; Stachenfeld et al., 2021; Kohl et al., 2024; Brandstetter et al., 2022b; Fanaskov et al., 2023; Bergamin et al., 2024; Sun et al., 2023; Hsieh et al., 2019; Tran et al., 2023; Li et al., 2023; Bonev et al., 2023). Nonetheless, the relative effectiveness of these strategies remains largely ambiguous, and transparent, systematic comparisons remain elusive.

Here, we systematically investigate the utility of symmetry constraints and physical conservation laws, alone and in combination. While both have proven useful for some tasks and architectures, to date there have been practically no systematic evaluations of their combination. Given the deep connections between conservation laws and PDE symmetries in physics (Noether, 1918), it is not clear *a priori* whether these constraints would prove redundant, or combine usefully for training PDE surrogates. Across multiple tasks, accuracy measures, architectures, training techniques, and scenarios, we show a clear, reproducible and robust benefit from these constraints for long rollout accuracy and generalization performance, and that they combine synergistically. To make them broadly applicable, we introduce novel input and output layers that extend these inductive biases to staggered grids for the first time.[1]

---

[1]Code is available at https://github.com/m-dml/double-constraint-pde-surrogates.

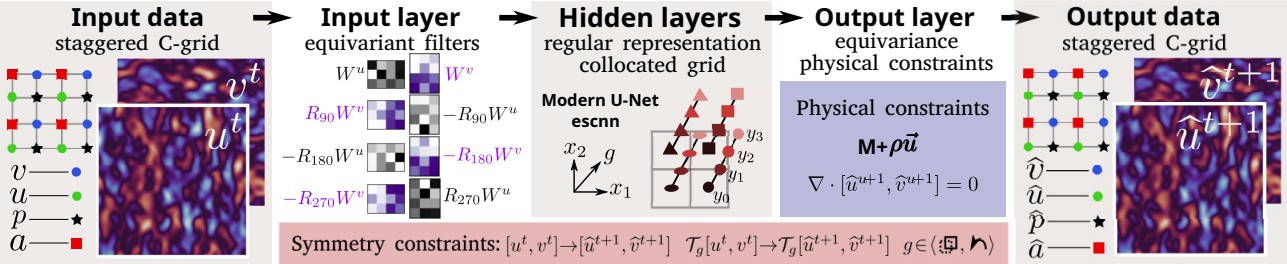

Figure 1. Symmetry- and physics-constrained neural surrogate for incompressible flow on a staggered grid. A rotation-equivariant input layer maps velocities onto a non-staggered regular representation, hidden layers employ steerable convolutions and the equivariant output layer enforces conservation laws on mass and momentum ($\mathbf{M} + \rho\vec{\mathbf{u}}$) as it maps to staggered velocities.

## 2. Background and Related Work

**Neural PDE surrogates** We aim to train neural networks to predict the time evolution of a system of PDEs. We consider time-dependent variable fields $\boldsymbol{w}(t, x) \in \mathbb{R}^m$, for $x \in \Omega \subset \mathbb{R}^d, t \in [0, T]$ and

$$\frac{\partial \boldsymbol{w}}{\partial t} = \mathcal{F}(t, \boldsymbol{x}, \boldsymbol{w}, \nabla\boldsymbol{w}, \nabla^2\boldsymbol{w}, \dots) \quad (1)$$

Starting from initial conditions (ICs) $\boldsymbol{w}(0, \boldsymbol{x})$ and boundary conditions (BCs) $B[\boldsymbol{w}](t, \boldsymbol{x}) = 0, \forall x \in \partial\Omega$, the solution can be advanced with a fixed time step:

$$\boldsymbol{w}(t + \Delta t, \cdot) = \mathcal{G}[\boldsymbol{w}(t, \cdot)], \quad (2)$$

where $\mathcal{G}$ is an update operator. To provide training data and evaluate performance we use a reference solution generated by a numerical solver with space- and time-discretized variable fields.

Recent studies have trained neural surrogates to approximate $\mathcal{G}$ (Greenfeld et al., 2019; Gupta & Brandstetter, 2023; List et al., 2024; Lippe et al., 2024; Li et al., 2021; Tripura & Chakraborty, 2023; Raonic et al., 2024). The neural network can also be combined with a numerical solver, in so-called 'hybrid methods' (Bar-Sinai et al., 2019; Tompson et al., 2017; Kochkov et al., 2021; Bukka et al., 2021; Long et al., 2019).

Training neural surrogates to remain stable and accurate over long autoregressive rollouts remains challenging. Several techniques have been proposed, including physical constraints, symmetry constraints, training with input noise, unrolled training and generative modeling. However, a clear consensus on the relative effectiveness of these approaches remains elusive, and their application to new tasks is not always straightforward.

**Symmetry equivariance** Suppose $f : \boldsymbol{w} \to \boldsymbol{z}$ is an operator mapping between two multidimensional variable fields $\boldsymbol{w}(\boldsymbol{x}), \boldsymbol{z}(\boldsymbol{x})$ defined on $\Omega \subset \mathbb{R}^d$. Then for a group $G$ of invertible transformations on $\mathbb{R}^d$, $f$ is *equivariant* if it commutes with the actions of $G$ on $\boldsymbol{w}$ and $\boldsymbol{z}$. Concretely, there should exist transformations $\mathcal{T}_g, \mathcal{T}'_g$ operating on $\boldsymbol{w}, \boldsymbol{z}$ respectively, such that

$$[f \circ \mathcal{T}_g \boldsymbol{w}](\boldsymbol{x}) = [\mathcal{T}'_g \circ f\boldsymbol{w}](\boldsymbol{x}), \quad \forall g \in G, \boldsymbol{x} \in \Omega \quad (3)$$

That is, transforming the inputs of $f$ will transform its outputs correspondingly. When $w$ is a scalar field, $\mathcal{T}$ and $\mathcal{T}'$ simply resample $w$ at coordinates defined by the action of $G$ on $\mathbb{R}^d$:

$$[T_g^{\text{scalar}} w](\boldsymbol{x}) = w(g^{-1}\boldsymbol{x}) \quad (4)$$

Other field types transform in more complex ways. For example, the action of a 90° rotation $R$ on a 2D vector field both resamples the field and rotates each vector:

$$[\mathcal{T}_R^{\text{vector}}(w_1, w_2)](\boldsymbol{x}) = (-w_2(R^{-1}\boldsymbol{x}), w_1(R^{-1}\boldsymbol{x})) \quad (5)$$

The range of possible actions is described by $G$'s group representations. Efficient, full-featured software packages exist for equivariant convolutions (Cesa et al., 2022) and self-attention (Romero & Cordonnier, 2021), and have proven useful in image classification (Chidester et al., 2019) and segmentation (Veeling et al., 2018), and to improve neural PDE surrogates (Wang et al., 2021; Helwig et al., 2023; Smets et al., 2023; Huang & Greenberg, 2023; Ruhe et al., 2024). Numerical integration methods can also benefit from maintaining PDE symmetries (Rebelo & Valiquette, 2013). We restrict ourselves to the discrete symmetry groups that hold precisely on regular grids, though some approaches have been proposed for the continuous symmetries that hold on the original PDEs (Weiler & Cesa, 2019; Thomas et al., 2018; Cesa et al., 2022; Knigge et al., 2024; Horie & Mitsume, 2022; Brandstetter et al., 2022a; Gasteiger et al., 2020; Lino et al., 2022; Toshev et al., 2023).

Non-equivariant architectures can also be trained using *data augmentation*, with $g \in G$ randomly sampled to transform input-target pairs during training (Brandstetter et al., 2022b). But while equivariant surrogates maintain symmetries precisely for any training or testing data, data augmentation results in imprecise equivariance that may fail to generalize beyond the training data.

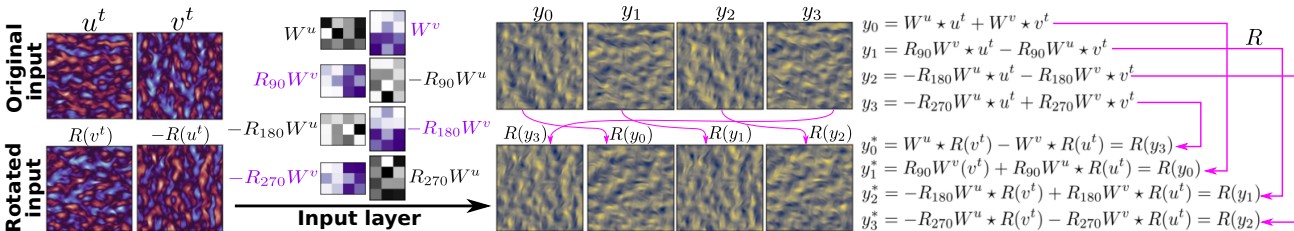

*Figure 2.* Action of rotation-equivariant input layer on staggered velocity fields (top left). The filter bank is transformed by each $g \in G$ to compute a $G$-indexed regular representation $y$. Rotation-transforming inputs (bottom left) yields permuted, rotated output channels.

**Staggered grids** Fluid dynamical systems are often simulated using staggered grids (Fig. 8), in which variables such as pressure, density, divergence or velocity along each axis are represented at different locations. This approach can avoid grid-scale numerical artifacts in numerical integration, and is widely used in fluid dynamics (Holl & Thuerey, 2024; Kochkov et al., 2021; Jasak, 2009; Stone et al., 2020) as well as atmospheric (Jungclaus et al., 2022; De Pondeca et al., 2011; Korn et al., 2022), and ocean models (Korn et al., 2022; Madec et al., 2023a;b). Staggered grids are common in in finite volume solvers, which generally respect conservation laws and offer other numerical advantages (Ferziger et al., 2019). However, current software implementations of equivariant network layers (Cesa et al., 2022; Romero & Cordonnier, 2021) cannot be applied to PDE variable fields on equivariant grids. This is because they assume that variables fields are all located at the same points, allowing the action of symmetries on these fields to be broken into two steps: a resampling step $x \rightarrow g^{-1}x$ carried out on the grid itself, and a transformation step $w \rightarrow \rho_g(w)$ carried out on PDE field variables $w \in \mathbb{R}^m$ at each point single grid point. This leads to overall transformations $w \rightarrow \mathcal{T}_g w$, such that $[\mathcal{T}_g w](x) = \rho_g(w(g^{-1}x))$. This is a valid assumption for PDEs on continuous spatial domains (eq. 5) or for collocated grids (eq. 1 of Weiler & Cesa 2019). But for staggered grids, the PDE fields are not represented as a vector of values $w(x)$ at each grid point $x$. Instead, each field is defined at different locations, which may be grid cell centers, interfaces or vertices. Thus, the spatial transformation of the grid and the transformation of local field values cannot be disentangled. Applying existing equivariant network layers to staggered PDE fields therefore breaks symmetry constraints.

Unfortunately, existing equivariant network layers (Cesa et al., 2022; Romero & Cordonnier, 2021) assume $\mathcal{T}_g$ can be described by a resampling operation followed by an independent transformation at each grid point as in Equation 5, but on staggered grids rotation and reflection do not take this form.

**Physical constraints** Neural surrogates have frequently been applied to physical systems, many of which contain known conservation laws. To improve accuracy, stability, and generalization capabilities, these laws can be imposed through additional loss terms (Read et al., 2019; Wang et al., 2021; Stachenfeld et al., 2021; Sorourifar et al., 2023). Taking the strategy of physics-derived loss terms to its ultimate limit, one arrives at unsupervised training on PDE-derived losses for discretized (Wandel et al., 2021; Michelis & Katzschmann, 2022) or continuous solutions (Raissi et al., 2019). Alternatively, one can reparameterize network outputs to respect hard constraints (Mohan et al., 2020; Beucler et al., 2021; Chalapathi et al., 2024; Cranmer et al., 2020; Greydanus et al., 2019). Here we focus on hard-constrained supervised learning with space- and time-discretized grids, which has proven more competitive in larger and more complex PDE systems (Takamoto et al., 2022).

## 3. Symmetry- and Physics-Constrained Neural Surrogates

In this work, we assess the separate and combined benefits of symmetries and conservation laws for neural PDE surrogates. To achieve this, we construct equivariant input layers that support staggered grids (Fig. 8), as well as output layers enforcing both equivariance and conservation laws. When comparing to non-equivariant networks, we replace equivariant convolutions using standard convolutions with the same size and padding options (Appendix B), adjusting channel width to match total parameter counts.

Fig. 1 demonstrates our overall framework for constructing equivariant, conservative neural surrogates. As an illustrative example, we show the incompressible Navier Stokes equations, with equivariance in translation and rotation, momentum conservation and a divergence-free condition (equivalent to mass conservation). Input data defined on staggered grids are mapped through novel equivariant input layers to a set of convolutional output channels defined at grid cell centers. Each internal activations consist of *regular representations*: groups of channels indexed by $G$,[2] on which $G$ acts by transforming each spatial field and by permuting the channels according to the group action (Cohen & Welling, 2016; Cesa et al., 2022). Essentially, regular representations are real-valued functions of the discrete symmetry

---

[2] Technically, by the non-translational subgroup of $G$.

group $G$. This formulation allows us to use the preexisting library `escnn` (Cesa et al., 2022) for all internal linear transformations between hidden layers. Finally, we employ novel output layers to map the regular representation back to the staggered grid while enforcing conservation laws as hard constraints.

**Input layers** We consider input data on staggered Arakawa C-grids (Fig. 8b) with square cells, and variable fields defined at cell centers (typically scalar fields such as pressure, surface height, or divergence), at cell interface midpoints (such as velocity components), and at vertices (such as vector potentials). For a $n \times n$ 2D grid of cells, there is a $(n+1) \times n$ grid of interfaces in the $x_1$ direction (along rows, including boundaries), and a $n \times (n+1)$ grid of interfaces in the $x_2$ direction (along columns).

We designed convolutional input layers to take scalar inputs at cell centers and/or vector fields with components defined at interfaces. Inputs at interfaces are first processed with a bank of convolutional filters, each of even size along the coordinate axis orthogonal to a single set of interfaces, and of odd size along all other axes (Fig. 2, left). This filter bank is *collectively* transformed according to each element of the symmetry group $G$, while being applied to the input data. Note that, similar to the transformation of vector fields (Eq. 5), these filter banks undergo collective transformation by rotations and reflections, not only through resampling, but also through permutation and sign flips (Fig. 2, right). When we rotation-transform input vector fields (Eq. 5), this has the effect of permuting and rotating the outputs of our input layer, as required for an equivariant mapping onto a regular representation (Fig. 2, magenta arrows), which the proof can be found in the Appendix C.2. Inputs at cell centers are processed with separate, standard equivariant convolution layers. Convolutions for both interface and center-defined input variables produce regular representation outputs, which are then combined to compute the total input to the network's first hidden layer. We provide implementations of 2D input layers for translation-rotation (p4) and translation-rotation-reflection (p4m). Further details on input layers can be found in the Appendix C.

**Output layers** We designed convolutional output layers mapping from regular representations to staggered C-grid variables (Fig. 8). As for the input layers, we use separate convolutional filter banks for cell- and interface-centered variables, but now additionally support vertex-centered scalar outputs for the purpose of enforcing physical constraints (see below). Scalar face-centered outputs are computed using pooling layers over a regular representation (Cohen & Welling, 2016). Vector field outputs at each cell interface are computed as linear combinations of regular representations at the two surrounding cell centers, with

constraints imposed on the weights to satisfy the equivariant transformation of vector fields (Eq. 5, details of output layers and their proofs in D). Vertex-centered scalar outputs are computed using even-sized square filters, followed by pooling layers operating over $G$-indexed channels.

**Conservation laws** We impose 3 types of conservation laws as hard constraints. For scalar quantities such as fluid surface height $\zeta$, we subtract the global mean of $\zeta^{t+1} - \zeta^t$ at each time step. For vector fields, we subtract the mean of each velocity component. As mass conservation in incompressible flows is equivalent to divergence-free velocity fields, we impose this by learning a vector potential $a$ defined at grid vertices, and compute velocities at grid cell interfaces as the curl $\nabla \times a$ to satisfy both mass and momentum conservation (Wandel et al., 2021). Further details and discussion of alternative approaches are found in appendix F.

*Table 1.* Geometric and physical constraints for SWEs

| Conservation laws | Symmetries | | |
|---|---|---|---|
| | ⊡ | ⊡↻ | ⊡↻⟋ |
| None ∅ | **p1/∅** | **p4/∅** | **p4m/∅** |
| Mass **M** | **p1/M** | **p4/M** | **p4m/M** |

*Table 2.* Geometric and physical constraints for INS

| Conservation laws | Symmetries | | |
|---|---|---|---|
| | ⊡ | ⊡↻ | ⊡↻⟋ |
| None ∅ | **p1/∅** | **p4/∅** | **p4m/∅** |
| Momentum $\rho\vec{u}$ | **p1/$\rho\vec{u}$** | **p4/$\rho\vec{u}$** | **p4m/$\rho\vec{u}$** |
| Mass/momentum **M**+$\rho\vec{u}$ | **p1/M**+$\rho\vec{u}$ | **p4/M**+$\rho\vec{u}$ | **p4m/M**+$\rho\vec{u}$ |

### 3.1. Surrogate Architectures

To measure the efficacy of symmetries and physical constraints we chose a flexible base architecture with efficient training and inference that has produced highly competitive results: the "modern U-net" (Gupta & Brandstetter, 2023), which modifies the original U-net (Ronneberger et al., 2015) for improved performance as a PDE surrogate (Appendix H). This architecture has shown strong results in (Kohl et al., 2024), and a similar version performed well in (Lippe et al., 2024). We used it without self-attention layers, which did not significantly affect our results. When constructing symmetry-respecting versions of the U-net, we confirmed equivariance held to numerical precision, but only if input/output layers for staggered grids were used (Appendix E).

In some experiments, we also compared to additional baselines. The Dilated ResNet architecture (drnet) has has performed well as a PDE surrogate (Stachenfeld et al., 2021), and we considered constrained and unconstrained versions. The rotation-equivariant U-net of Wang et al., 2021 was also enforces hard symmetry constraints, but was not intended for staggered grids. Fourier neural operators (FNOs) com-

bine local operations with filtering in frequency space (Li et al., 2021), and we use unconstrained versions.

## 3.2. Training

We trained neural surrogates using a MSE loss $\mathcal{L} = \frac{1}{N} \left\| \widehat{\boldsymbol{w}}^{t+1} - \boldsymbol{w}^{t+1} \right\|_2^2$, where $N$ is the number of discretized PDE field values. All data fields were normalized by subtracting the mean and dividing by the standard deviation, with common values for both components of vector fields. We trained on 2 A100 GPUs with the ADAM optimizer (Kingma, 2014), batch size 32 and initial learning rate 1e-4. We employed early stopping when validation loss did not reduce for 10 epochs, and accepted network weights with the best validation loss throughout the training process.

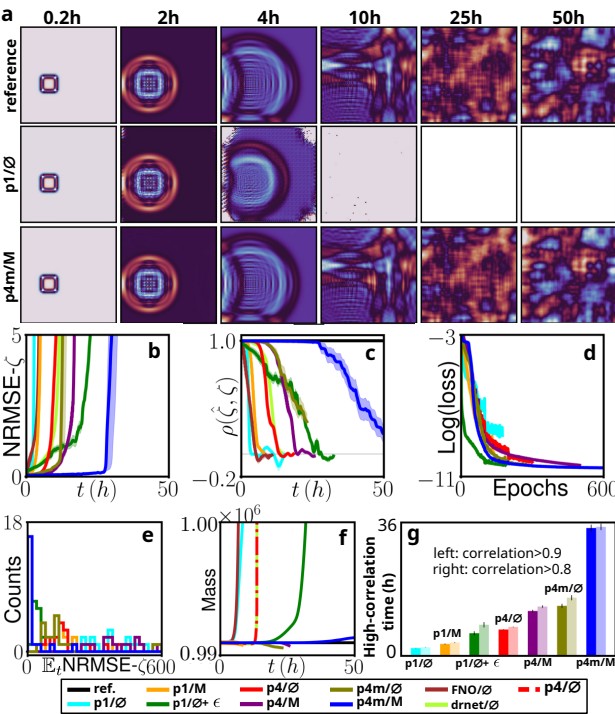

*Figure 3.* p4m/M (symmetry+physics constraints) outperforms other networks with similar parameter counts on SWEs. (a) Reference surface disturbance $\zeta$ with predictions from p1/∅ and p4m/M. (b-c) Accuracy over 50h rollouts, with standard error of the mean over 20 ICs. (d) Training loss over iterations. (e) Histogram of $\mathbb{E}_t$NRMSE over 20 ICs. (f) Violation of mass conservation for all methods (black line shows reference simulation). (g) High correlation times for each model.

## 4. PDE Systems

We considered two challenging 2D fluid dynamic PDEs, with the same staggered grid and symmetries but different variables, BCs/ICs, reference solvers and conservation laws. Full sets of constraints for each system and names for each combination appear in Tables (1-2), while PDE parameters and further numerical details appear in Tables (6-7).

## 4.1. Closed Shallow Water System

The shallow water equations (SWEs) are widely used to describe a quasi-static motion in a homogeneous incompressible fluid with a free surface. We consider nonlinear SWEs in momentum- and mass conservative form on the domain $\Omega$ with 'closed' Dirichlet BCs (Song et al., 2018):

$$\frac{\partial \boldsymbol{u}}{\partial t} = -C_D \frac{1}{h} \boldsymbol{u} |\boldsymbol{u}| - g\nabla\zeta + a_h \nabla^2 \boldsymbol{u} \tag{6}$$

$$\frac{\partial \zeta}{\partial t} = -\nabla \cdot (h\boldsymbol{u}) \tag{7}$$

$$\boldsymbol{u} = \boldsymbol{0}, \qquad\qquad \text{on } \partial\Omega \tag{8}$$

where $\zeta$ is fluid surface elevation, $\boldsymbol{u} = [u, v]$ is the velocity field, $d$ and $h$ respectively represent the undisturbed- and disturbed fluid depth (so that $h = d + \zeta$) and $\partial\Omega$ is a closed domain boundary. $a_h$ is the horizontal turbulent momentum exchange coefficient, $C_D$ is the bottom drag coefficient and $g$ is gravitational acceleration. SWEs simulations exhibit travelling waves that reflect from domain boundaries, temporarily increasing in height as they self-collide. This system is more challenging than previous SWE tasks with open (Takamoto et al., 2022) or periodic BCs (Gupta & Brandstetter, 2023), due to the combination of self-interfering wave patterns, incompressibility and altered dynamics at pixels near the domain boundaries.

**Numerical reference solution** Closed BCs and incompressibility lead to stiff dynamics, so explicit solvers are inefficient. Instead, we generate data using a semi-implicit scheme (Backhaus, 1983) that represents $\zeta$ and $[u, v]$ on a staggered Arakawa C-grid (Arakawa, 1977) and solves a sparse linear system at each time step $\Delta t = 300s$.

Grids are $100 \times 100$, $100 \times 99$, and $99 \times 100$ respectively for $\zeta$, $u$, and $v$. We trained on 50 simulations spanning 50 h (600 time steps) each. ICs were $\zeta = 0$ except for a 0.1 m high square-shaped elevation, and $[u, v] = 0$. The square had side length uniformly distributed from 2-28 grid cells and random position. The solver was implemented in Fortran and required 67 s/IC on a compute node with 48 CPUs. Testing and validation data included 10 simulations. Surrogates used the solver's time step. Since the time evolution of this SWE systems depends on the location of boundaries, we provide a binary boundary mask to the network as an additional input field with scalar values defined at grid cell centers. We note that this binary mask is invariant to rotations and reflections.

**Symmetries and conservation laws** The shallow water system in Eqs. 7-8 is equivariant to rotations and reflections. Solver equivariance was empirically verified in Fig. 9. The only conserved quantity for SWE is mass (defined as $\Delta x^2 h$ times fluid density, so that the mean of $\zeta$ is also conserved). Momentum is not conserved in this SWE system, and an

*Table 3.* NRMSE-$\zeta$, $\rho(\hat{\zeta}, \zeta)$, and average absolute mass and total energy errors for SWEs surrogates at 1h and 25h. NaN values indicate some rollouts diverged to infinity.

| Model | NRMSE-$\zeta$ | | $\rho(\hat{\zeta}, \zeta)$ | | Mean(\|Mass-ref.\|) | | Mean(\|Total energy-ref.\|) | |
|---|---|---|---|---|---|---|---|---|
| | 1h | 25h | 1h | 25h | 1h | 25h | 1h | 25h |
| FNO/∅ | 0.58±0.03 | NaN | 0.8391±0.0143 | NaN | 22.09±0.19 | NaN | 86592±748 | NaN |
| drnet/∅ | 0.11±0.01 | NaN | 0.9977±0.0006 | NaN | 4.84±0.01 | NaN | 18572±24 | NaN |
| p1/∅ | 0.14 ±0.02 | NaN | 0.9957 ±0.0012 | NaN | 11.7±0.09 | NaN | 45939±366 | NaN |
| p1/M | 0.10±0.01 | NaN | 0.9934±0.0016 | NaN | 0.06 ±0.01 | NaN | 128.0 ±23.9 | NaN |
| p4/∅ | 0.035±0.003 | NaN | 0.9992±2e-4 | NaN | 0.17±0.03 | NaN | 768.0±112.7 | NaN |
| p4/M | 0.034±0.004 | 1743±386 | 0.9992±2e-4 | -0.02±0.01 | 0.04 ±0.01 | 223.4 ±42.4 | 153.6±17.2 | 2.6e9±1.9e8 |
| p4m/∅ | 0.045±0.005 | NaN | 0.9993±2e-4 | NaN | 2.95 ±0.01 | NaN | 11500±48 | NaN |
| p4m/M | **0.032±0.004** | **0.14±0.02** | **0.9993±2e-4** | **0.987±4e-3** | **0.03 ±0.01** | **0.29 ±0.03** | **121.6±19.1** | **1094±181** |

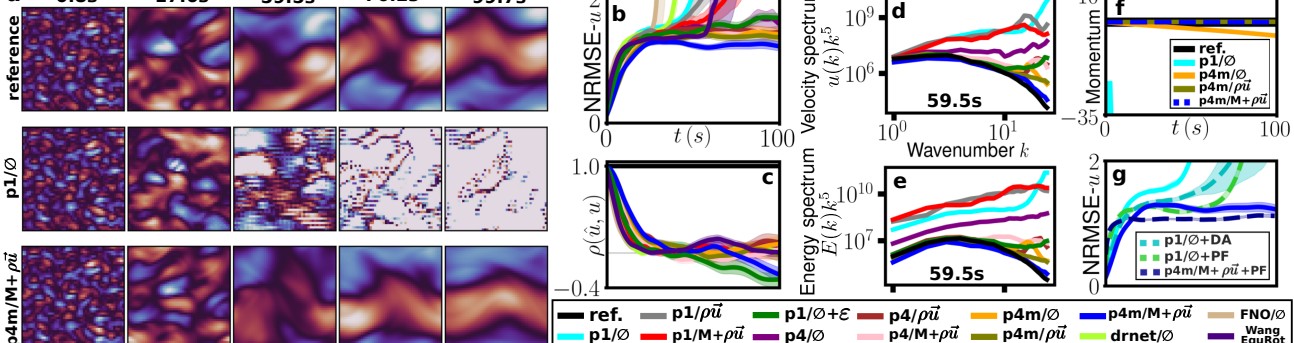

*Figure 4.* p4m/M+$\rho\vec{u}$ outperforms other networks with similar parameter counts on INS. (a) Reference horizontal velocity with predictions from p1/∅ and p4m/M+$\rho\vec{u}$. (b-c) Accuracy over 50h rollouts, with standard error of the mean over 30 ICs. (d-e) Log-log plots of the average velocity power spectrum and energy spectrum from 30 ICs at $59.5s$. Spectra measure the strength of the chaotic field's features for each wavenumber $k$ (number of cycles across the domain). Both the velocity and energy spectra p4m/M+$\rho\vec{u}$ align best with the reference. Spectra are scaled by $k^5$. (f) The momentum during the entire evolution time for p1/∅ and all p4m models. (g) Comparison of data-augmentation (**DA**) and pushforward trick (**PF**).

*Table 4.* The NRMSE-$u$ of decaying turbulence at 36s and 60s for Modern U-net, drnet, Modern U-net with pushforward trick (**PF**) and Modern U-net with data augmentation (**DA**).

| Model | Modern U-Net | | Dilated ResNet | | Modern U-net + PF | | Modern U-net + DA | |
|---|---|---|---|---|---|---|---|---|
| | 36s | 60s | 36s | 60s | 36s | 60s | 36s | 60s |
| p1/∅ | 1.56 ±0.05 | 3.32 ±0.12 | 1.57 ±0.04 | 2.3e4 ±1.9e4 | 1.88 ±0.03 | 12.03 ±0.42 | 1.36 ±0.03 | 1.59 ±0.12 |
| p1/$\rho\vec{u}$ | 1.50±0.04 | 2.16±0.22 | 1.24±0.02 | 1.48±0.25 | 1.38 ±0.04 | 1.47 ±0.07 | 1.36 ±0.03 | 1.42 ±0.07 |
| p1/M+$\rho\vec{u}$ | 1.46±0.05 | 1.71±0.15 | 1.33±0.04 | 1.53±0.05 | 1.43 ±0.02 | 1.52 ±0.03 | 1.31 ±0.03 | 1.38 ±0.06 |
| p4/∅ | 1.45±0.04 | 1.49±0.05 | 1.34±0.04 | 1.42±0.05 | 1.52 ±0.03 | 2.33 ±0.07 | 1.39 ±0.04 | 1.48 ±0.07 |
| p4/$\rho\vec{u}$ | 1.46±0.03 | 1.50±0.07 | 1.33±0.04 | 1.53±0.06 | 1.49 ±0.03 | 1.54 ±0.04 | 1.35 ±0.03 | 1.41 ±0.06 |
| p4/M+$\rho\vec{u}$ | 1.38±0.03 | 1.41±0.05 | 1.35±0.04 | 1.47±0.04 | 1.43 ±0.02 | 1.54 ±0.04 | 1.34 ±0.04 | 1.31 ±0.04 |
| p4m/∅ | 1.37±0.04 | 1.42±0.05 | 1.36±0.04 | 1.47±0.07 | 1.66 ±0.05 | 1.65 ±0.05 | 1.38 ±0.04 | 1.39 ±0.07 |
| p4m/$\rho\vec{u}$ | 1.33±0.03 | 1.40±0.06 | 1.29±0.04 | 1.40±0.06 | 1.32 ±0.03 | 1.37 ±0.03 | 1.35 ±0.03 | 1.40 ±0.06 |
| p4m/M+$\rho\vec{u}$ | **1.29±0.04** | **1.20±0.05** | **1.29±0.03** | **1.30±0.07** | **1.07 ±0.01** | **1.06 ±0.02** | **1.24 ±0.03** | **1.25 ±0.03** |

eastward wave will reverse and westwards after reflecting from a boundary. In reality, this would be compensated by a slight change in the momentum of the Earth, but this is not simulated.

### 4.2. Decaying Turbulence

The incompressible Navier–Stokes equations (INS) describe momentum balance for incompressible Newtonian fluids.

Our 2D version relates velocities $\boldsymbol{u} = [u, v]$ to pressure $p$:

$$\frac{\partial \boldsymbol{u}}{\partial t} + (\boldsymbol{u} \cdot \nabla)\boldsymbol{u} = -\frac{\nabla p}{\rho} + \mu \nabla^2 \boldsymbol{u} \tag{9}$$

$$\nabla \cdot \boldsymbol{u} = 0 \tag{10}$$

where $\rho$ is fluid density and $\mu$ is kinematic viscosity. Here we consider the 'decaying turbulence' scenario introduced by (Kochkov et al., 2021). The velocity field is initialized as filtered Gaussian noise containing high spatial frequencies. Predicting the evolution of the velocity field is challenging, since eddy size and Reynolds number change over time as structures in the flow field coalesce, and the velocity field becomes smoother and more uniform over time.

**Numerical reference solution** We solve Eqs. 9-10 with C-grid staggering of velocities, using `jax-cfd` (Kochkov et al., 2021). We follow previous data generation procedures (Kochkov et al., 2021; Stachenfeld et al., 2021), with a $576 \times 576$ grid and $44$ ms time step over 224 seconds. Training data were coarsened to a time step of $0.84$ s, and resolution was reduced to $48 \times 48$ (Stachenfeld et al., 2021) using face-averaging to conserve momentum and the divergence-free condition. A burn-in of 148 coarsened steps leaves 120 steps for training. We trained on 100 ICs consisting of filtered Gaussian noise with peak spectral density at wavenumber 10 (that is, 10 cycles across the spatial domain). We used 10 initial conditions for testing and validation.

**Symmetries and conservation laws** We empirically verified the INS solver's equivariance (Fig. 10). Conserved quantities include momentum (equivalent to a constant mean velocities since $\rho$ is constant), and mass (through the divergence-free condition).

# 5. Results

## 5.1. Closed Shallow Water System

We first trained and evaluated neural surrogates for the SWE task. We followed a hybrid learning strategy, based on the observation that the semiimplicit numerical integration scheme calculates $\zeta^{t+1}$ slowly with an iterative solver, but then calculates $[u^{t+1}, v^{t+1}]$ given $\zeta^{t+1}$ quickly through a mathematical formula. We therefore trained surrogates to predict only $\widehat{\zeta}^{t+1}$, and calculated $[\widehat{u}^{t+1}, \widehat{v}^{t+1}]$ as in the numerical solver (Appendix I). Keeping parameter counts constant, we compared networks equivariant to 3 symmetry groups: p1 (translation only, as in standard CNNs), p4 (translation-rotation) and p4m (translation-rotation-reflection). We also compared mass conserving networks (M) to those without physical constraints ($\varnothing$). Table 1 lists all constraint combinations used for training, which took 0.5 h for non-equivariant networks and 2h for equivariant networks on 2 A100 GPUs. Table 3 shows surrogate

accuracy, along with errors in mass and total energy.

Fig. 3a compares autoregressive rollouts from unconstrained (p1/$\varnothing$) and maximally constrained networks (p4m/M). p4m/M maintained accurate results for a much greater time interval, and in this case was visually indistinguishable from the reference solution throughout the simulation (for other surrogates, see Fig. 15). Over 20 random held-out ICs, p4m/M exhibited lower normalized RMSE values and higher correlations than other networks (Figs. 3b-c)). p4m/M also outperformed p1/$\varnothing$ trained with input noise (Stachenfeld et al., 2021; Lippe et al., 2024), and unconstrained FNO and drnet architectures. Compared to other networks, p4m/M trained for more epochs before early stopping occurred, reached a lower validation loss (Fig. 3d) and produced accurate results for a greater fraction of held-out ICs (Fig. 3e). Mass conservation was respected up to numerical precision by the original solver and physics-constrained architectures, but not by other surrogates (Fig. 3f). Overall, we found that symmetry constraints were more effective than conservation laws, but that equivariant surrogates could be further improved by physical constraints.

## 5.2. Decaying Turbulence

We next trained and evaluated neural surrogates for INS. Here we used the velocity fields $[u, v]$ as both inputs and outputs. As for SWEs, we tested p1, p4 and p4m equivariance, but now considered 3 levels of physical constraints: unconstrained ($\varnothing$), momentum conservation ($\rho\vec{u}$) and mass/momentum conservation (M+$\rho\vec{u}$). Table (2) lists all constraint combinations used for training, which took 0.4 h for nonequivariant networks and 1.4 h for equivariant networks on 2 A100 GPUs. Table 4 lists accuracy after 36s and 60s for all surrogates.

Figure (4-a) compares autoregressive rollouts from unconstrained (p1/$\varnothing$) and maximally constrained networks (p4m/M+$\rho\vec{u}$). As for the SWEs, both constraint types improved accuracy and stability of INS surrogates (Fig. 4b-c), and double constraints were best, also outperforming networks trained with input noise (Stachenfeld et al., 2021). Unconstrained networks were particularly susceptible to numerical instability in this task (rollouts in Fig. 18-19).

**Spectral consistency** To evaluate the performance of neural surrogates beyond the time at which their predictions decorrelate from the reference solution, we followed previous studies (Kochkov et al., 2021; Lippe et al., 2024; Stachenfeld et al., 2021) in further comparing the power spectra of predicted velocity fields, and of energy fields $\frac{1}{2}|\vec{u}|^2$, to those of the reference solver. Even after average correlation with the reference solution reached 0, we found that p4m/M+$\rho\vec{u}$ networks matched the spectra of the reference solver better than other methods, consistently across multiple rollout times and especially at the highest spatial frequencies

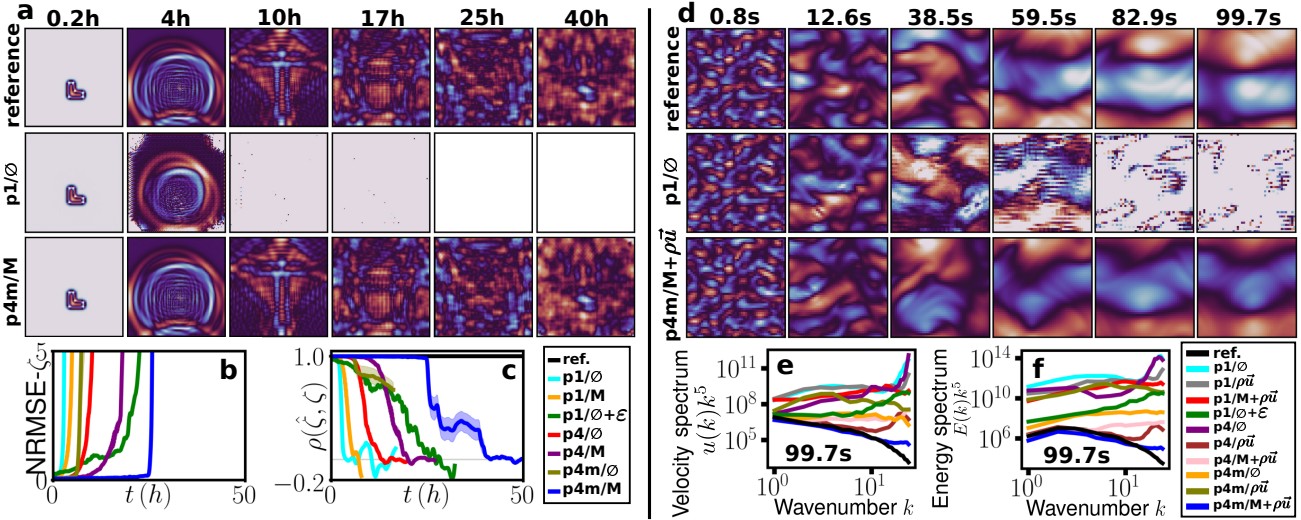

*Figure 5.* Generalization beyond training data. (a) SWE rollouts from p1/∅ p4m/M on L-shaped ICs. (b-c) Accuracy of each network over six generalization tests (Appendix J.11). (d) INS rollouts from p1/∅ and p4m/M+$\rho\vec{u}$ on ICs with peak wavenumber 8. (e-f) Velocity- and energy spectra for INS at $t = 99.7$ s, averaged over 10 ICs.

*Table 5.* Comparison of nRMSE for zonal velocity, energy spectrum error (ESE), and $\rho(\hat{u}, u)$ for real ocean currents predicted 12h and 120h ahead. Architecture and hyperparameters for Equ$_{rot}$ Unet are as described in (Wang et al., 2021)

| Model | 12h | | | 120h | | |
|---|---|---|---|---|---|---|
| | nRMSE | ESE | $\rho(\hat{u}, u)$ | nRMSE | ESE | $\rho(\hat{u}, u)$ |
| p1/∅ | 1.52±0.04 | 5.33±6.23 | 0.01±0.02 | 1.66±0.03 | 13.9±11.4 | -0.02±0.02 |
| Equ$_{rot}$ Unet (Wang et al., 2021) | 0.91±0.02 | 0.85±0.16 | 0.31±0.02 | 1.14±0.03 | 1.07±1.11 | 0.00±0.02 |
| p4m/M+$\rho\vec{u}$ | **0.88±0.02** | **0.80±0.12** | **0.45±0.03** | **1.10±0.03** | **0.94±0.90** | **0.11±0.02** |

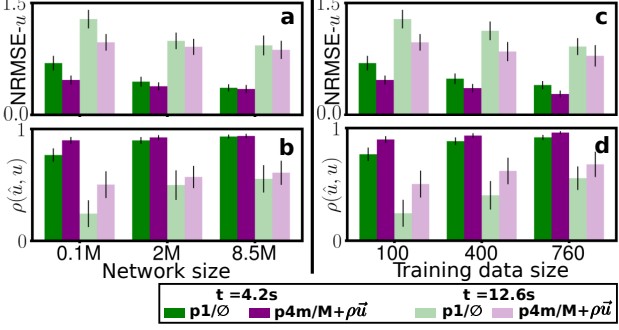

*Figure 6.* Accuracy of symmetry- and physics-constrained INS models across data and network sizes, at 4.2 and 12.6 s. (a-b) NRMSE-$u$ and $\rho(\hat{u}, u)$ vs. network size for p1/∅ and p4m/M+$\rho\vec{u}$. (c-d) NRMSE-$u$ and $\rho(\hat{u}, u)$ for p1/∅ and p4m/M+$\rho\vec{u}$ vs. training datasets size.

(Figs. 4d-e, additional spectra in Fig. 21). Conservation of momentum was respected only by momentum-constrained networks (Fig. 4f, 20). Training p4m/M+$\rho\vec{u}$ surrogates with input noise resulted in lower accuracy but excellent long-term numerical stability (Fig. 22).

**Specialized training modes** We further compared the performance gains from symmetries and physical constraints to those offered by specialized training modes for PDE sur-

rogates (Fig. 4g): data augmentation and the pushforward trick. We applied data augmentation using the p4m symmetry group, such that velocity fields on the staggered C-grid were transformed consistently with the numerical solver: $S \circ \mathcal{T}_g(w) = \mathcal{T}_g \circ S$. We applied the pushforward trick as in (Brandstetter et al., 2022c), with the MSE loss computed after two autoregressive time steps, but gradients backpropagated only one step. Both of these training modes improved performance compared to standard training of p1/∅, but could not match the accuracy of p4m/M+$\rho\vec{u}$ with standard training (Fig. 4g, Table 4). Data augmentation had, as expected, no effect on p4m surrogates, but pushforward training of p4m/M+$\rho\vec{u}$ produced the most accurate surrogate overall, showing that doubly constrained networks benefit from autoregressive training. We present further results and details on these modes in appendices J.9-J.10.

**Alternate base architecture** Beyond evaluating the utility of constraining modern U-nets, we also evaluated symmetries and physical constraints for Dilated ResNet surrogates of INS(Table 4). Performance was strikingly consistent with previous results, with symmetries more effective than physical constraints but further benefits observable when combining both. Thus, the separate and combined effects of our constraints were consistent across these two architectures.

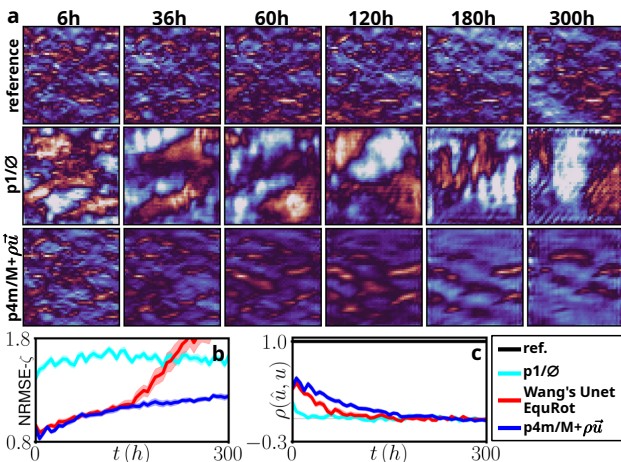

*Figure 7.* Prediction of ocean current observations by p1/∅, Wang's Equ$_\text{rot}$ (Unet) and p4m/M+$\rho\vec{u}$. (a) Zonal ocean currents from reference observations and predictions. (b-c) NRMSE-u and correlation for forecasts up to 300 h ahead (averaged over 30 ICs).

We present additional results for drnets in Appendix J.8.

### 5.3. Generalization to Novel ICs

**Closed Shallow Water System** Fig. 5a shows an initial condition consisting of an 'L'-shaped elevation (details in Fig. 31). Randomly varying the elevation's location and shape, we found p4m/M to outperform alternatives with equal parameter counts (Fig. 5b-c). Additional results are shown in Fig. 29-30.

**Decaying Turbulence** We tested surrogates on ICs with peak wavenumber changed from 10 to 8. p4m/M+$\rho\vec{u}$ matched the reference solver (Fig. 5d) and its spectra (Figs. 5e-f) most closely. Additional results are shown in Figs. 32-34.

### 5.4. Effects of Network and Dataset Size

To investigate how the benefits of symmetries and physical constraints scale with network and dataset size, we trained p1/∅ and p4m/M+$\rho\vec{u}$ networks with 0.1M, 2M and and 8.5M parameters on 100 INS simulations. At both 4.2 and 12.6 s we observed lower errors and high correlations for p4m/M+$\rho\vec{u}$ for all network sizes. The relative advantage of p4m/M+$\rho\vec{u}$ over p1/∅ was greatest for smaller networks and longer forecast horizons, and overall performance was best for larger networks. CPU and GPU inference speeds are reported in Appendix J.13.

Training 0.1M-parameter p1/∅ and p4m/M+$\rho\vec{u}$ networks on 100, 400 and 760 simulations showed constraints enhance performance robustly across dataset size (Fig. 6c-d). Improvements were greater on larger datasets and longer rollouts (additional results and spectra in Fig. 35).

### 5.5. Predicting Ocean Current Observations

We trained neural surrogates to predict real-world observations of ocean currents (Wang et al., 2021) (data and training details in Appendix J.14). Doubly-constrained p4m/M+$\rho\vec{u}$ predicted future observations better than p1/∅, and also outperformed the equivariant network proposed in Wang et al., 2021, in terms of NRMSE and correlation (Fig. 7) as well as Energy Spectrum Error (ESE, Table 5).

## 6. Discussion

We enforced hard constraints on symmetries and conservation laws for neural PDE surrogates. We extended the applicability of previous techniques to staggered grids, and systematically tested performance across tasks and constraints. Symmetries were more effective, but conservation laws were not redundant. Double constraints matched reference simulations, individually and statistically, better than multiple architectures, pushforward training, and data augmentation. In our challenging PDE tasks, insufficiently constrained surrogates diverged towards infinity before completing their rollouts (Table 3), as did the PDEs' conserved quantities themselves in some cases (Fig. 3f, 4f). These results underline physics- and symmetry-constrained surrogates as a promising strategy when long-term accuracy is required but data are expensive or limited, as in weather, climate and industrial fluid mechanics.

**Limitations & Future work** For large enough networks and datasets, constraints might be learned from the data (Stachenfeld et al., 2021; Watt-Meyer et al., 2023), but the benefit of constraints grows with rollout length even for large networks and datasets. Thus, constraints are likely relevant for longer time scales, e.g. for seasonal forecasting and climate projections (Kochkov et al., 2024; Watt-Meyer et al., 2023; Nguyen et al., 2023). How constraints limit error accumulation remains unclear, but empirical investigations of how error accumulation correlates with constraint violations over time and ICs could provide some clarity.

We considered mass and momentum conservation, and symmetries of square 2D grids. Future work could pursue other PDEs such as hyperbolic equations (Takamoto et al., 2022), energy conservation (Cranmer et al., 2020), local vs. global conservation (McGreivy & Hakim, 2023), continuous symmetry groups (Cohen et al., 2018; Esteves et al., 2018), alternative grids and meshes (Cohen et al., 2019; De Haan et al., 2021; Horie & Mitsume, 2022), generalization to new geometries (Wandel et al., 2021; Horie & Mitsume, 2022), unrolled training (Brandstetter et al., 2022c; List et al., 2024), invariant measure learning (Schiff et al., 2024) and generative modeling (Lippe et al., 2024; Kohl et al., 2024).

## Acknowledgements

We thank A.C. Bekar and N. Kumar for comments on the manuscript.

## Impact Statement

This paper presents work whose goal is to advance the field of Machine Learning. There are many potential societal consequences of our work, none which we feel must be specifically highlighted here.

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

## A. Symmetries of SWEs and INS with C-grid staggering

### A.1. Grid Discretizations

The Arakawa C-grid (Arakawa, 1977) is a discretization technique frequently employed in numerical simulation, particularly in fluid dynamics. In this section, the C-grid staggering for shallow water equations (SWEs) and incompressible Navier–Stokes equations (INS) is presented (Figure 8).

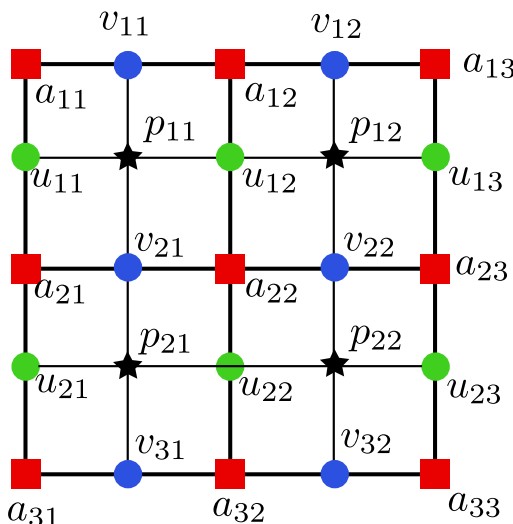

*Figure 8.* Arakawa's C-grid staggering approach has been employed for time integrating the of shallow water equations (SWEs) and incompressible Navier–Stokes equations (INS). (a) Red square points denote the vertical displacement of the free surface, denoted by $\zeta$. The circular points correspond to the velocities $u$ and $v$. (b) In the discretization of the INS system, the square points represents the vector potential $a$, while the velocity is indicated by the circular points. The black stars indicate pressure, which is calculated by the numerical solver at each time step but not used as a prognostic variable to solve for velocity in future time steps.

### A.2. The symmetries of shallow water equations

The shallow water system with closed boundary is represented by Equations (6-8). For a specified set of PDE fields (i.e., $\zeta$, $u$, and $v$) at a given time $t$, the solution of the equation for $\zeta$ at the subsequent time interval can be expressed as $S_\zeta(\zeta, u, v)$. The symmetry transformations (flip, rotation, and flip-rotation) of SWEs with the numerical solver $S$ can be expressed as follows:

$$\textbf{flip}: \quad S_\zeta(F(\zeta), F(u), -F(v)) = F(S_\zeta(\zeta, u, v)) \tag{11}$$

$$\textbf{rotation}: \quad S_\zeta(R(\zeta), -R(v), R(u)) = R(S_\zeta(\zeta, u, v)) \tag{12}$$

$$\textbf{flip} - \textbf{rotation}: \quad S_\zeta(R(F(\zeta)), R(F(v)), R(F(u))) = R(F(S_\zeta(\zeta, u, v))) \tag{13}$$

In this section, the numerical solution of the equations of motion for the variable $\zeta$ in the subsequent time step will be denoted by $S_\zeta$. The actions of the flipping operator, $F$, and the rotation operator, $R$, will also be defined for these PDE fields. Figure 9 show the equivariance of the numerical solver to these transformations.

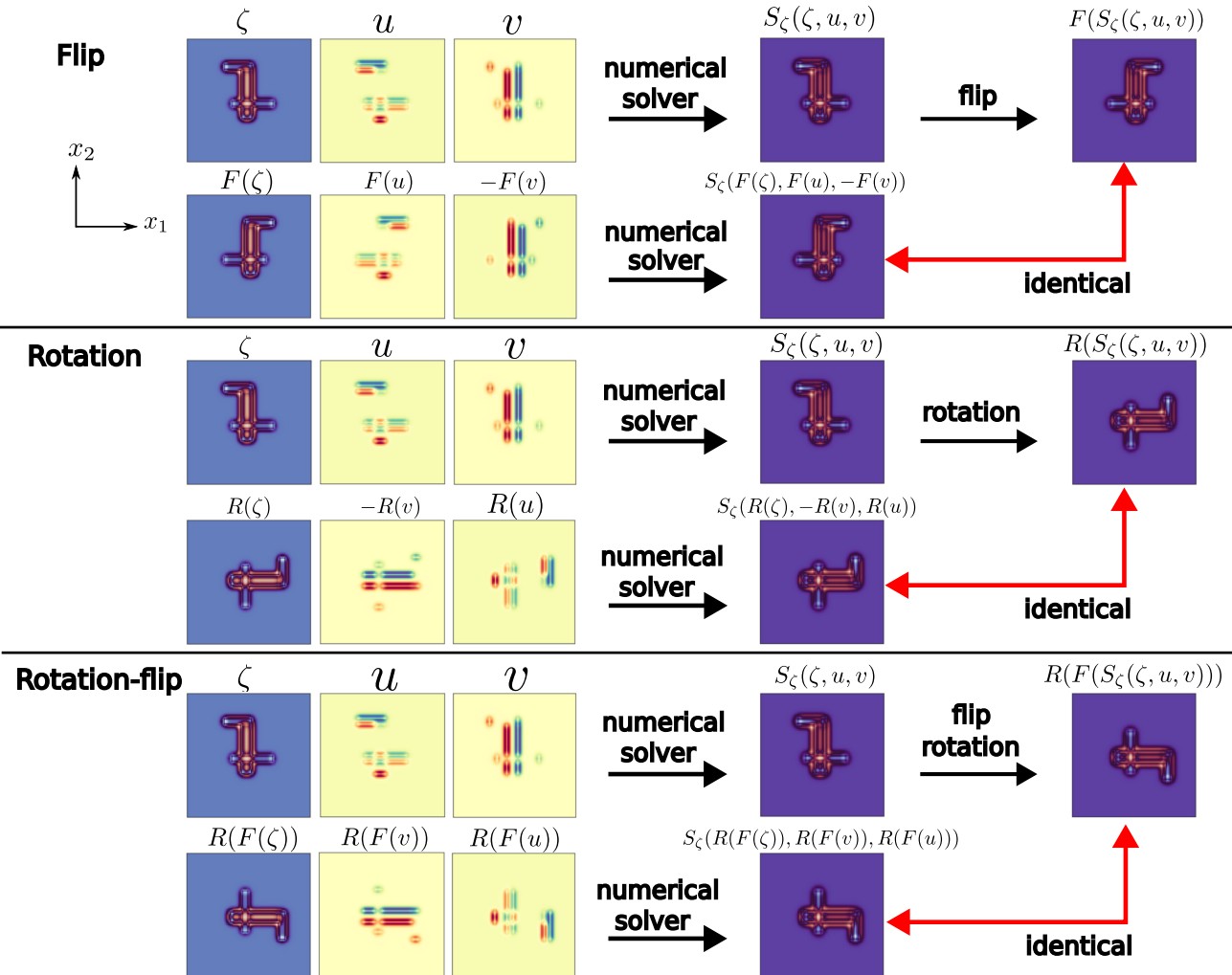

*Figure 9.* Empirical validation of the symmetries of the numerical SWE solver is demonstrated through three transformations: flip, rotation, and flip-rotation. These plots correspond to Equations (11-13).

### A.3. Symmetries of the incompressible Navier–Stokes equations

The incompressible Navier–Stokes equations are presented in Equations (10-9). Given a set of initial conditions $u$ and $v$ at a particular time $t$, the solutions to the velocity equation are expressed as $S_u(u,v)$ and $S_v(u,v)$ at time $t+1$. The mathematical description of the flip, rotation, and flip-rotation symmetries of INS is as follows:

$$\textbf{flip}: \begin{cases} S_u(-F(u), F(v)) = -F(S_u(u,v)) \\ S_v(-F(u), F(v)) = F(S_v(u,v)) \end{cases} \tag{14}$$

$$\textbf{rotation}: \begin{cases} S_u(R(v), -R(u)) = R(S_v(u,v)) \\ S_v(R(v), -R(u)) = R(-S_u(u,v)) \end{cases} \tag{15}$$

$$\textbf{flip} - \textbf{rotation}: \begin{cases} S_u(R(F(v)), -R(-F(u))) = R(F(S_v(u,v))) \\ S_v(R(F(v)), -R(-F(u))) = -R(-F(S_u(u,v))) \end{cases} \tag{16}$$

As shown in Figure 10, the symmetries of the incompressible Navier–Stokes equations encompass are generated by flipping and rotation.

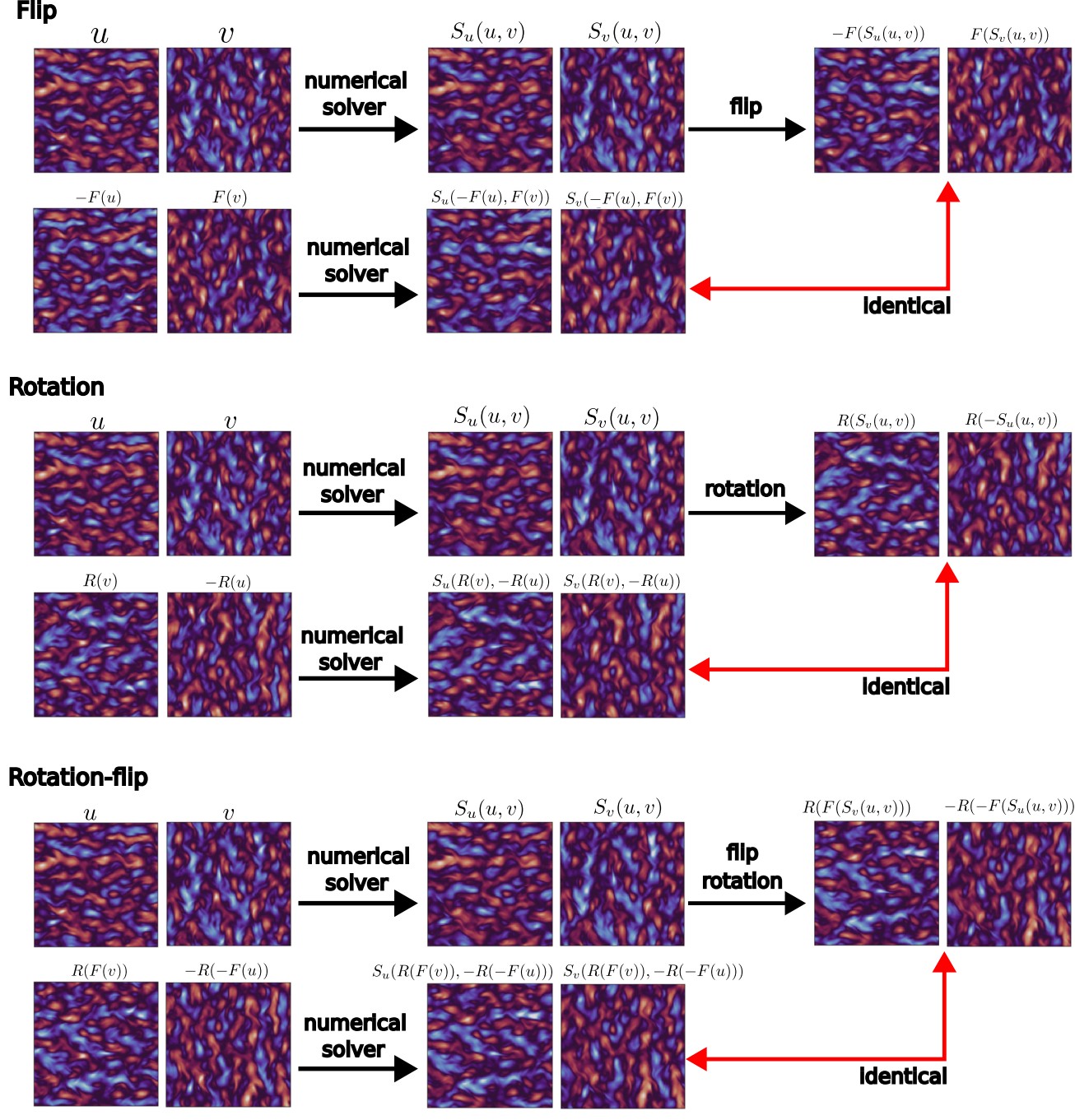

*Figure 10.* Empirical confirmation of the equivariance of the INS solver.

## B. Padding options

In certain numerical solvers, although a C-grid staggering is employed, the software generates output of an identical size for each component of the vector field. This phenomenon necessitates a thorough examination of the selected conventions for padding and boundary representation in the outputs. In such instances, the implementation of a padding technique becomes imperative to restore the vector field to its standard C-grid staggering. One potential solution is the implementation of periodic padding for the periodic boundary conditions (BCs). It is imperative to emphasize that the physical properties of the data, such as the divergence in the incompressible Navier–Stokes equations and the BCs, must remain unaltered during

the implementation of this correction.

## C. Group equivariant input layers for SWE and INS

In this section, we present explicit formulas for equivariant input layers and show that they satisfy equivariance relations. For the sake of brevity, we have elected to include only the proof for the p4 input layer for SWEs, but extending this proof to p4m or INS is straightforward.

### C.1. Group equivariant input layer for SWEs

Given that the input data for shallow water equations (SWEs) utilizes C-grid staggering, as illustrated in Figure 8, it is necessary to construct an input layer that aligns with the C-grid staggering while preserving equivariance. On the C-grid, the velocity components $u$ and $v$ possess distinct dimensions. To address this, we propose the utilization of two rectangular filters, denoted as $W^u_{j,i,\cdot,\cdot}$ and $W^v_{j,i,\cdot,\cdot}$, for the variables $u$ and $v$, respectively. The symbol $\cdot$ is employed to denote all values along a given axis. The filter $W$ is a $c_{\text{in}} \times c_0 \times K \times S$ array, where $c_0$ denotes the batch size, $c_{\text{in}}$ signifies the number of input channels, and $K \times S$ represents the filter size.

For instance, on a periodic 2D grid tiling the torus with $3 \times 3$ grid cells, the sizes are $4 \times 3$ for $u$ and $3 \times 4$ for $v$. While the height field $\zeta$ is defined at grid cell centers, $u, v$ are defined on the interfaces (edges) between the grid cells. In the context of performing group transformations in the input layer, it is imperative to exchange the filters for $u$ and $v$ to ensure compatibility with the dimensions of the input variables.

**p4** We first introduce an input layer of the p4 group transformation for SWEs, which has four channels obtained from four different rotated filters. The output $y$ of this input layer is defined as follows ($\star$ denotes convolution):

$$y^1_{j,0,\cdot,\cdot} = \sum_{i=0}^{c^\zeta_{in}-1} \left( W^\zeta_{j,i,\cdot,\cdot} \star \zeta_{i,\cdot,\cdot} \right) + \sum_{i=0}^{c^u_{in}-1} \left( W^u_{j,i,\cdot,\cdot} \star u_{i,\cdot,\cdot} \right) + \sum_{i=0}^{c^v_{in}-1} \left( W^v_{j,i,\cdot,\cdot} \star v_{i,\cdot,\cdot} \right) + b_j, \tag{17}$$

$$y^1_{j,1,\cdot,\cdot} = \sum_{i=0}^{c^\zeta_{in}-1} \left( R^{90^\circ}_{\text{rot}}(W^\zeta_{j,i,\cdot,\cdot}) \star \zeta_{i,\cdot,\cdot} \right) + \sum_{i=0}^{c^u_{in}-1} \left( -R^{90^\circ}_{\text{rot}}(W^v_{j,i,\cdot,\cdot}) \star u_{i,\cdot,\cdot} \right)$$
$$+ \sum_{i=0}^{c^v_{in}-1} \left( R^{90^\circ}_{\text{rot}}(W^u_{j,i,\cdot,\cdot}) \star v_{i,\cdot,\cdot} \right) + b_j, \tag{18}$$

$$y^1_{j,2,\cdot,\cdot} = \sum_{i=0}^{c^\zeta_{in}-1} \left( R^{180^\circ}_{\text{rot}}(W^\zeta_{j,i,\cdot,\cdot}) \star \zeta_{i,\cdot,\cdot} \right) + \sum_{i=0}^{c^u_{in}-1} \left( -R^{180^\circ}_{\text{rot}}(W^u_{j,i,\cdot,\cdot}) \star u_{i,\cdot,\cdot} \right)$$
$$+ \sum_{i=0}^{c^v_{in}-1} \left( -R^{180^\circ}_{\text{rot}}(W^v_{j,i,\cdot,\cdot}) \star v_{i,\cdot,\cdot} \right) + b_j, \tag{19}$$

$$y^1_{j,3,\cdot,\cdot} = \sum_{i=0}^{c^\zeta_{in}-1} \left( R^{270^\circ}_{\text{rot}}(W^\zeta_{j,i,\cdot,\cdot}) \star \zeta_{i,\cdot,\cdot} \right) + \sum_{i=0}^{c^u_{in}-1} \left( R^{270^\circ}_{\text{rot}}(W^v_{j,i,\cdot,\cdot}) \star u_{i,\cdot,\cdot} \right)$$
$$+ \sum_{i=0}^{c^v_{in}-1} \left( -R^{270^\circ}_{\text{rot}}(W^u_{j,i,\cdot,\cdot}) \star v_{i,\cdot,\cdot} \right) + b_j. \tag{20}$$

where $W^\zeta_{j,i,\cdot,\cdot}$ is a square filter for $\zeta$, for example $4 \times 4$. $b$ is a $c_{\text{out}}$-element vector.

**p4m** Next, we define an input layer for p4m that uses the same logic as the p4 input layer. It has 8 different group transformations, including flips and rotations.

$$y_{j,0,\cdot,\cdot}^1 = \sum_{i=0}^{c_{in}^\zeta-1} \left( W_{j,i,\cdot,\cdot}^\zeta \star \zeta_{i,\cdot,\cdot} \right) + \sum_{i=0}^{c_{in}^u-1} \left( W_{j,i,\cdot,\cdot}^u \star u_{i,\cdot,\cdot} \right) + \sum_{i=0}^{c_{in}^v-1} \left( W_{j,i,\cdot,\cdot}^v \star v_{i,\cdot,\cdot} \right) + b_j, \tag{21}$$

$$\begin{aligned}
y_{j,1,\cdot,\cdot}^1 = \sum_{i=0}^{c_{in}^\zeta-1} \left( F_{\text{flip}}(W_{j,i,\cdot,\cdot}^\zeta) \star \zeta_{i,\cdot,\cdot} \right) + \sum_{i=0}^{c_{in}^u-1} \left( F_{\text{flip}}(W_{j,i,\cdot,\cdot}^u) \star u_{i,\cdot,\cdot} \right) \\
+ \sum_{i=0}^{c_{in}^v-1} \left( F_{\text{flip}}(W_{j,i,\cdot,\cdot}^v) \star v_{i,\cdot,\cdot} \right) + b_j,
\end{aligned} \tag{22}$$

$$\begin{aligned}
y_{j,2,\cdot,\cdot}^1 = \sum_{i=0}^{c_{in}^\zeta-1} \left( R_{\text{rot}}^{90°}(W_{j,i,\cdot,\cdot}^\zeta) \star \zeta_{i,\cdot,\cdot} \right) + \sum_{i=0}^{c_{in}^u-1} \left( R_{\text{rot}}^{90°}(W_{j,i,\cdot,\cdot}^v) \star u_{i,\cdot,\cdot} \right) \\
+ \sum_{i=0}^{c_{in}^v-1} \left( R_{\text{rot}}^{90°}(W_{j,i,\cdot,\cdot}^u) \star v_{i,\cdot,\cdot} \right) + b_j,
\end{aligned} \tag{23}$$

$$\begin{aligned}
y_{j,3,\cdot,\cdot}^1 = \sum_{i=0}^{c_{in}^\zeta-1} \left( F_{\text{flip}}(R_{\text{rot}}^{90°}(W_{j,i,\cdot,\cdot}^\zeta)) \star \zeta_{i,\cdot,\cdot} \right) + \sum_{i=0}^{c_{in}^u-1} \left( F_{\text{flip}}(R_{\text{rot}}^{90°}(W_{j,i,\cdot,\cdot}^v)) \star u_{i,\cdot,\cdot} \right) \\
+ \sum_{i=0}^{c_{in}^v-1} \left( F_{\text{flip}}(R_{\text{rot}}^{90°}(W_{j,i,\cdot,\cdot}^u)) \star v_{i,\cdot,\cdot} \right) + b_j,
\end{aligned} \tag{24}$$

$$\begin{aligned}
y_{j,4,\cdot,\cdot}^1 = \sum_{i=0}^{c_{in}^\zeta-1} \left( R_{\text{rot}}^{180°}(W_{j,i,\cdot,\cdot}^\zeta) \star \zeta_{i,\cdot,\cdot} \right) + \sum_{i=0}^{c_{in}^u-1} \left( R_{\text{rot}}^{180°}(W_{j,i,\cdot,\cdot}^u) \star u_{i,\cdot,\cdot} \right) \\
+ \sum_{i=0}^{c_{in}^v-1} \left( R_{\text{rot}}^{180°}(W_{j,i,\cdot,\cdot}^v) \star v_{i,\cdot,\cdot} \right) + b_j,
\end{aligned} \tag{25}$$

$$\begin{aligned}
y_{j,5,\cdot,\cdot}^1 = \sum_{i=0}^{c_{in}^\zeta-1} \left( F_{\text{flip}}(R_{\text{rot}}^{180°}(W_{j,i,\cdot,\cdot}^\zeta)) \star \zeta_{i,\cdot,\cdot} \right) + \sum_{i=0}^{c_{in}^u-1} \left( F_{\text{flip}}(R_{\text{rot}}^{180°}(W_{j,i,\cdot,\cdot}^u)) \star u_{i,\cdot,\cdot} \right) \\
+ \sum_{i=0}^{c_{in}^v-1} \left( F_{\text{flip}}(R_{\text{rot}}^{180°}(W_{j,i,\cdot,\cdot}^v)) \star v_{i,\cdot,\cdot} \right) + b_j,
\end{aligned} \tag{26}$$

$$\begin{aligned}
y_{j,6,\cdot,\cdot}^1 = \sum_{i=0}^{c_{in}^\zeta-1} \left( R_{\text{rot}}^{270°}(W_{j,i,\cdot,\cdot}^\zeta) \star \zeta_{i,\cdot,\cdot} \right) + \sum_{i=0}^{c_{in}^u-1} \left( R_{\text{rot}}^{270°}(W_{j,i,\cdot,\cdot}^v) \star u_{i,\cdot,\cdot} \right) \\
+ \sum_{i=0}^{c_{in}^v-1} \left( R_{\text{rot}}^{270°}(W_{j,i,\cdot,\cdot}^u) \star v_{i,\cdot,\cdot} \right) + b_j,
\end{aligned} \tag{27}$$

$$\begin{aligned}
y_{j,7,\cdot,\cdot}^1 = \sum_{i=0}^{c_{in}^\zeta-1} \left( F_{\text{flip}}(R_{\text{rot}}^{270°}(W_{j,i,\cdot,\cdot}^\zeta)) \star \zeta_{i,\cdot,\cdot} \right) + \sum_{i=0}^{c_{in}^u-1} \left( F_{\text{flip}}(R_{\text{rot}}^{270°}(W_{j,i,\cdot,\cdot}^v)) \star u_{i,\cdot,\cdot} \right) \\
+ \sum_{i=0}^{c_{in}^v-1} \left( F_{\text{flip}}(R_{\text{rot}}^{270°}(W_{j,i,\cdot,\cdot}^u)) \star v_{i,\cdot,\cdot} \right) + b_j,
\end{aligned} \tag{28}$$

where the filters $W_{j,i,\cdot,\cdot}^u$ and $W_{j,i,\cdot,\cdot}^v$ are rectangles and the filter $W_{j,i,\cdot,\cdot}^\zeta$ is a square.

## C.2. Proof of p4 equivariance for the SWE p4 input layer

Here we prove group equivariance (Cohen & Welling, 2016) for the SWE p4 input layer. To this end, the following equation must be demonstrated: $y(R(\zeta), -R(v), R(u)) = R(y(\zeta, u, v))$ where $y$ is the output of the p4 input layer for SWEs. This equation is simply the rotation symmetry of a shallow water system as depicted in Equation 12.

The forms of the ordinary input layer, denoted by $y(\zeta, u, v)$ in Equations (17-20), have previously been described. Applying them to a rotated inputs consisting of $R_{\text{rot}}^{90°}(\zeta_{i,\cdot,\cdot})$, $-R_{\text{rot}}^{90°}(v_{i,\cdot,\cdot})$, and $R_{\text{rot}}^{90°}(u_{i,\cdot,\cdot})$ we get:

$$\tilde{y}_{j,0,\cdot,\cdot}^1 = \sum_{i=0}^{c_{in}^\zeta-1} \left( W_{j,i,\cdot,\cdot}^\zeta \star R_{\text{rot}}^{90°}(\zeta_{i,\cdot,\cdot}) \right) + \sum_{i=0}^{c_{in}^u-1} \left( W_{j,i,\cdot,\cdot}^u \star -R_{\text{rot}}^{90°}(v_{i,\cdot,\cdot}) \right)$$
$$+ \sum_{i=0}^{c_{in}^v-1} \left( W_{j,i,\cdot,\cdot}^v \star R_{\text{rot}}^{90°}(u_{i,\cdot,\cdot}) \right) + b_j, \tag{29}$$

$$\tilde{y}_{j,1,\cdot,\cdot}^1 = \sum_{i=0}^{c_{in}^\zeta-1} \left( R_{\text{rot}}^{90°}(W_{j,i,\cdot,\cdot}^\zeta) \star R_{\text{rot}}^{90°}(\zeta_{i,\cdot,\cdot}) \right) + \sum_{i=0}^{c_{in}^u-1} \left( -R_{\text{rot}}^{90°}(W_{j,i,\cdot,\cdot}^v) \star -R_{\text{rot}}^{90°}(v_{i,\cdot,\cdot}) \right)$$
$$+ \sum_{i=0}^{c_{in}^v-1} \left( R_{\text{rot}}^{90°}(W_{j,i,\cdot,\cdot}^u) \star R_{\text{rot}}^{90°}(u_{i,\cdot,\cdot}) \right) + b_j, \tag{30}$$

$$\tilde{y}_{j,2,\cdot,\cdot}^1 = \sum_{i=0}^{c_{in}^\zeta-1} \left( R_{\text{rot}}^{180°}(W_{j,i,\cdot,\cdot}^\zeta) \star R_{\text{rot}}^{90°}(\zeta_{i,\cdot,\cdot}) \right) + \sum_{i=0}^{c_{in}^u-1} \left( -R_{\text{rot}}^{180°}(W_{j,i,\cdot,\cdot}^u) \star -R_{\text{rot}}^{90°}(v_{i,\cdot,\cdot}) \right)$$
$$+ \sum_{i=0}^{c_{in}^v-1} \left( -R_{\text{rot}}^{180°}(W_{j,i,\cdot,\cdot}^v) \star R_{\text{rot}}^{90°}(u_{i,\cdot,\cdot}) \right) + b_j, \tag{31}$$

$$\tilde{y}_{j,3,\cdot,\cdot}^1 = \sum_{i=0}^{c_{in}^\zeta-1} \left( R_{\text{rot}}^{270°}(W_{j,i,\cdot,\cdot}^\zeta) \star R_{\text{rot}}^{90°}(\zeta_{i,\cdot,\cdot}) \right) + \sum_{i=0}^{c_{in}^u-1} \left( R_{\text{rot}}^{270°}(W_{j,i,\cdot,\cdot}^v) - R_{\text{rot}}^{90°}(v_{i,\cdot,\cdot}) \right)$$
$$+ \sum_{i=0}^{c_{in}^v-1} \left( -R_{\text{rot}}^{270°}(W_{j,i,\cdot,\cdot}^u) \star R_{\text{rot}}^{90°}(u_{i,\cdot,\cdot}) \right) + b_j. \tag{32}$$

Applying the same a 90-degree rotation to the regular representation output for *unrotated* inputs, we get:

$$
\begin{aligned}
R_{\text{rot}}^{90^\circ}(y_{j,0,\cdot,\cdot}^1) = R_{\text{rot}}^{90^\circ}\Bigg( & \sum_{i=0}^{c_{in}^\zeta - 1}\left(W_{j,i,\cdot,\cdot}^\zeta \star \zeta_{i,\cdot,\cdot}\right) + \sum_{i=0}^{c_{in}^u - 1}\left(W_{j,i,\cdot,\cdot}^u \star u_{i,\cdot,\cdot}\right) \\
& + \sum_{i=0}^{c_{in}^v - 1}\left(W_{j,i,\cdot,\cdot}^v \star v_{i,\cdot,\cdot}\right) + b_j\Bigg) = \tilde{y}_{j,1,\cdot,\cdot}^1
\end{aligned}
\tag{33}
$$

$$
\begin{aligned}
R_{\text{rot}}^{90^\circ}(y_{j,1,\cdot,\cdot}^1) = R_{\text{rot}}^{90^\circ}\Bigg( & \sum_{i=0}^{c_{in}^\zeta - 1}\left(R_{\text{rot}}^{90^\circ}(W_{j,i,\cdot,\cdot}^\zeta) \star \zeta_{i,\cdot,\cdot}\right) + \sum_{i=0}^{c_{in}^u - 1}\left(-R_{\text{rot}}^{90^\circ}(W_{j,i,\cdot,\cdot}^v) \star u_{i,\cdot,\cdot}\right) \\
& + \sum_{i=0}^{c_{in}^v - 1}\left(R_{\text{rot}}^{90^\circ}(W_{j,i,\cdot,\cdot}^u) \star v_{i,\cdot,\cdot}\right) + b_j\Bigg) = \tilde{y}_{j,2,\cdot,\cdot}^1
\end{aligned}
\tag{34}
$$

$$
\begin{aligned}
R_{\text{rot}}^{90^\circ}(y_{j,2,\cdot,\cdot}^1) = R_{\text{rot}}^{90^\circ}\Bigg( & \sum_{i=0}^{c_{in}^\zeta - 1}\left(R_{\text{rot}}^{180^\circ}(W_{j,i,\cdot,\cdot}^\zeta) \star \zeta_{i,\cdot,\cdot}\right) + \sum_{i=0}^{c_{in}^u - 1}\left(-R_{\text{rot}}^{180^\circ}(W_{j,i,\cdot,\cdot}^u) \star u_{i,\cdot,\cdot}\right) \\
& + \sum_{i=0}^{c_{in}^v - 1}\left(-R_{\text{rot}}^{180^\circ}(W_{j,i,\cdot,\cdot}^v) \star v_{i,\cdot,\cdot}\right) + b_j\Bigg) = \tilde{y}_{j,3,\cdot,\cdot}^1
\end{aligned}
\tag{35}
$$

$$
\begin{aligned}
R_{\text{rot}}^{90^\circ}(y_{j,3,\cdot,\cdot}^1) = R_{\text{rot}}^{90^\circ}\Bigg( & \sum_{i=0}^{c_{in}^\zeta - 1}\left(R_{\text{rot}}^{270^\circ}(W_{j,i,\cdot,\cdot}^\zeta) \star \zeta_{i,\cdot,\cdot}\right) + \sum_{i=0}^{c_{in}^u - 1}\left(R_{\text{rot}}^{270^\circ}(W_{j,i,\cdot,\cdot}^v) \star u_{i,\cdot,\cdot}\right) \\
& + \sum_{i=0}^{c_{in}^v - 1}\left(-R_{\text{rot}}^{270^\circ}(W_{j,i,\cdot,\cdot}^u) \star v_{i,\cdot,\cdot}\right) + b_j\Bigg) = \tilde{y}_{j,0,\cdot,\cdot}^1
\end{aligned}
\tag{36}
$$

Applying the definition of the rotation operator's action on the regular representation (rotation + permutation), we see the channels of these two regular representations match, so we arrive at the formula:

$$
R_{\text{rot}}^{90^\circ}(y^1(\zeta_{i,\cdot,\cdot}, u_{i,\cdot,\cdot}, v_{i,\cdot,\cdot})) = \tilde{y}^1(R_{\text{rot}}^{90^\circ}(\zeta_{i,\cdot,\cdot}), -R_{\text{rot}}^{90^\circ}(v_{i,\cdot,\cdot}), R_{\text{rot}}^{90^\circ}(u_{i,\cdot,\cdot}))
\tag{37}
$$

Thus we have proved the group equivariance of the p4 input layer in shallow water equations.

### C.3. Group equivariant input layer for INS

**p4** The system state for the incompressible Navier–Stokes equations consists of velocity fields $u$ and $v$, which possess disparate dimensions in the C-grid staggering. Consequently, two rectangular filters, denoted as $W_{j,i,\cdot,\cdot}^u$ and $W_{j,i,\cdot,\cdot}^v$, are required for the velocity field. According to the symmetries of rotation of INS in Equation (15), we first construct a p4 input layer for INS as follows:

$$
y_{j,0,\cdot,\cdot}^1 = \sum_{i=0}^{c_{in}^u - 1}\left(W_{j,i,\cdot,\cdot}^u \star u_{i,\cdot,\cdot}\right) + \sum_{i=0}^{c_{in}^v - 1}\left(W_{j,i,\cdot,\cdot}^v \star v_{i,\cdot,\cdot}\right) + b_j,
\tag{38}
$$

$$
y_{j,1,\cdot,\cdot}^1 = \sum_{i=0}^{c_{in}^u - 1}\left(R_{\text{rot}}^{90^\circ}(W_{j,i,\cdot,\cdot}^v) \star u_{i,\cdot,\cdot}\right) + \sum_{i=0}^{c_{in}^v - 1}\left(-R_{\text{rot}}^{90^\circ}(W_{j,i,\cdot,\cdot}^u) \star v_{i,\cdot,\cdot}\right) + b_j,
\tag{39}
$$

$$
y_{j,2,\cdot,\cdot}^1 = \sum_{i=0}^{c_{in}^u - 1}\left(-R_{\text{rot}}^{180^\circ}(W_{j,i,\cdot,\cdot}^u) \star u_{i,\cdot,\cdot}\right) + \sum_{i=0}^{c_{in}^v - 1}\left(-R_{\text{rot}}^{180^\circ}(W_{j,i,\cdot,\cdot}^v) \star v_{i,\cdot,\cdot}\right) + b_j,
\tag{40}
$$

$$
y_{j,3,\cdot,\cdot}^1 = \sum_{i=0}^{c_{in}^u - 1}\left(-R_{\text{rot}}^{270^\circ}(W_{j,i,\cdot,\cdot}^v) \star u_{i,\cdot,\cdot}\right) + \sum_{i=0}^{c_{in}^v - 1}\left(R_{\text{rot}}^{270^\circ}(W_{j,i,\cdot,\cdot}^u) \star v_{i,\cdot,\cdot}\right) + b_j.
\tag{41}
$$

**p4m** Subsequently, in accordance with the flip-rotation symmetries of INS in Equation (16), the p4m input layer is defined

by the following eight equations:

$$y_{j,0,\cdot,\cdot}^1 = \sum_{i=0}^{c_{in}^u-1} \left( W_{j,i,\cdot,\cdot}^u \star u_{i,\cdot,\cdot} \right) + \sum_{i=0}^{c_{in}^v-1} \left( W_{j,i,\cdot,\cdot}^v \star v_{i,\cdot,\cdot} \right) + b_j, \tag{42}$$

$$y_{j,1,\cdot,\cdot}^1 = \sum_{i=0}^{c_{in}^u-1} \left( F_{\text{flip}}(W_{j,i,\cdot,\cdot}^u) \star u_{i,\cdot,\cdot} \right) + \sum_{i=0}^{c_{in}^v-1} \left( F_{\text{flip}}(W_{j,i,\cdot,\cdot}^v) \star v_{i,\cdot,\cdot} \right) + b_j, \tag{43}$$

$$y_{j,2,\cdot,\cdot}^1 = \sum_{i=0}^{c_{in}^u-1} \left( R_{\text{rot}}^{90°}(W_{j,i,\cdot,\cdot}^v) \star u_{i,\cdot,\cdot} \right) + \sum_{i=0}^{c_{in}^v-1} \left( R_{\text{rot}}^{90°}(W_{j,i,\cdot,\cdot}^u) \star v_{i,\cdot,\cdot} \right) + b_j, \tag{44}$$

$$y_{j,3,\cdot,\cdot}^1 = \sum_{i=0}^{c_{in}^u-1} \left( F_{\text{flip}}(R_{\text{rot}}^{90°}(W_{j,i,\cdot,\cdot}^v)) \star u_{i,\cdot,\cdot} \right) + \sum_{i=0}^{c_{in}^v-1} \left( F_{\text{flip}}(R_{\text{rot}}^{90°}(W_{j,i,\cdot,\cdot}^u)) \star v_{i,\cdot,\cdot} \right) + b_j, \tag{45}$$

$$y_{j,4,\cdot,\cdot}^1 = \sum_{i=0}^{c_{in}^u-1} \left( R_{\text{rot}}^{180°}(W_{j,i,\cdot,\cdot}^u) \star u_{i,\cdot,\cdot} \right) + \sum_{i=0}^{c_{in}^v-1} \left( R_{\text{rot}}^{180°}(W_{j,i,\cdot,\cdot}^v) \star v_{i,\cdot,\cdot} \right) + b_j, \tag{46}$$

$$y_{j,5,\cdot,\cdot}^1 = \sum_{i=0}^{c_{in}^u-1} \left( F_{\text{flip}}(R_{\text{rot}}^{180°}(W_{j,i,\cdot,\cdot}^u)) \star u_{i,\cdot,\cdot} \right) + \sum_{i=0}^{c_{in}^v-1} \left( F_{\text{flip}}(R_{\text{rot}}^{180°}(W_{j,i,\cdot,\cdot}^v)) \star v_{i,\cdot,\cdot} \right) + b_j, \tag{47}$$

$$y_{j,6,\cdot,\cdot}^1 = \sum_{i=0}^{c_{in}^u-1} \left( R_{\text{rot}}^{270°}(W_{j,i,\cdot,\cdot}^v) \star u_{i,\cdot,\cdot} \right) + \sum_{i=0}^{c_{in}^v-1} \left( R_{\text{rot}}^{270°}(W_{j,i,\cdot,\cdot}^u) \star v_{i,\cdot,\cdot} \right) + b_j, \tag{48}$$

$$y_{j,7,\cdot,\cdot}^1 = \sum_{i=0}^{c_{in}^u-1} \left( F_{\text{flip}}(R_{\text{rot}}^{270°}(W_{j,i,\cdot,\cdot}^v)) \star u_{i,\cdot,\cdot} \right) + \sum_{i=0}^{c_{in}^v-1} \left( F_{\text{flip}}(R_{\text{rot}}^{270°}(W_{j,i,\cdot,\cdot}^u)) \star v_{i,\cdot,\cdot} \right) + b_j, \tag{49}$$

### C.4. Empirical validation of equivariance for SWE and INS input layers

In the preceding sections, the forms of group equivariant input layers for SWEs and INS were introduced. In this section, the equivariance of input layers for SWEs and INS is demonstrated empirically. Figure 11 illustrates the group equivariant input layer of SWEs, specifically the p4 and p4m layers. Given that the output of the input layer possesses four and eight group-indexed channels in the regular representations for p4 and p4m, respectively, an average is taken of the representations to produce a single equivariant scalar field output.

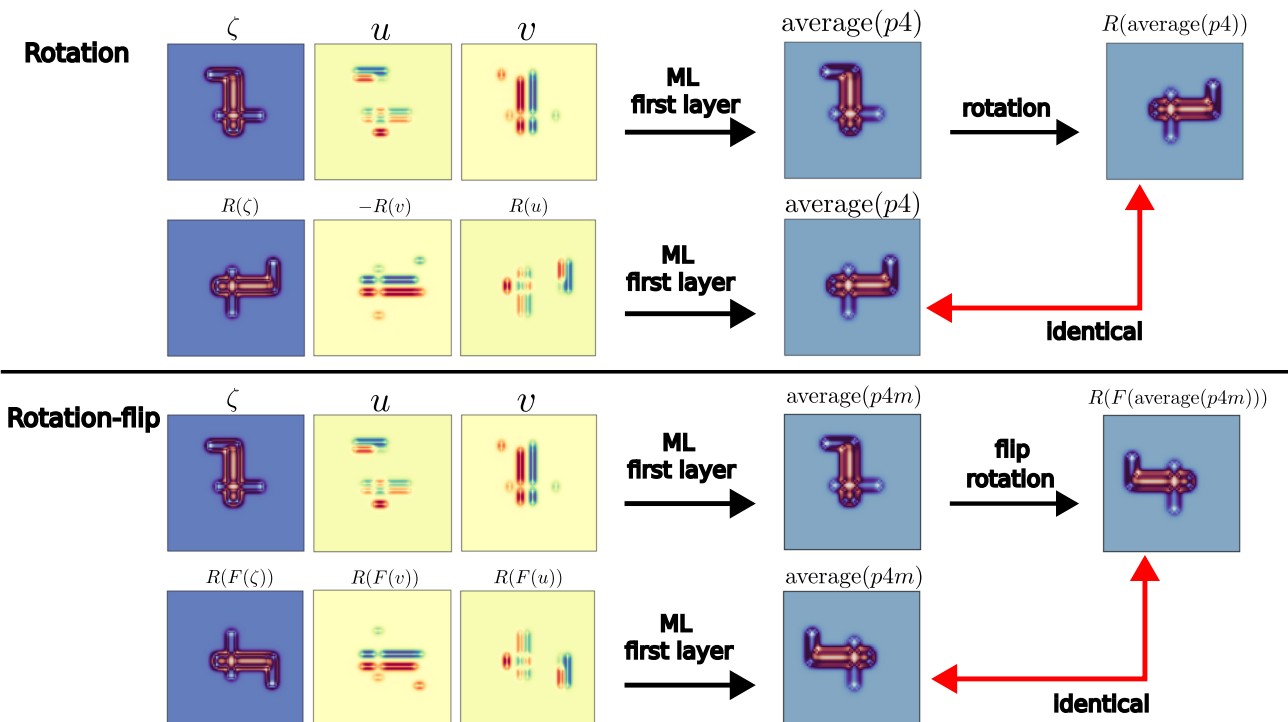

*Figure 11.* Example illustrating equivariance of the SWE input layer for groups p4 and p4m on staggered grid input data. The equivariance relation is observed to hold, to numerical precision. In this case, a $G$-averaged regular representation from the input layer's output is employed to visualize the equivariance.

Figure 12 illustrates the equivariance of p4 and p4m group equivariant input layers for INS. It can be seen that the input layer is equivariant in both cases.

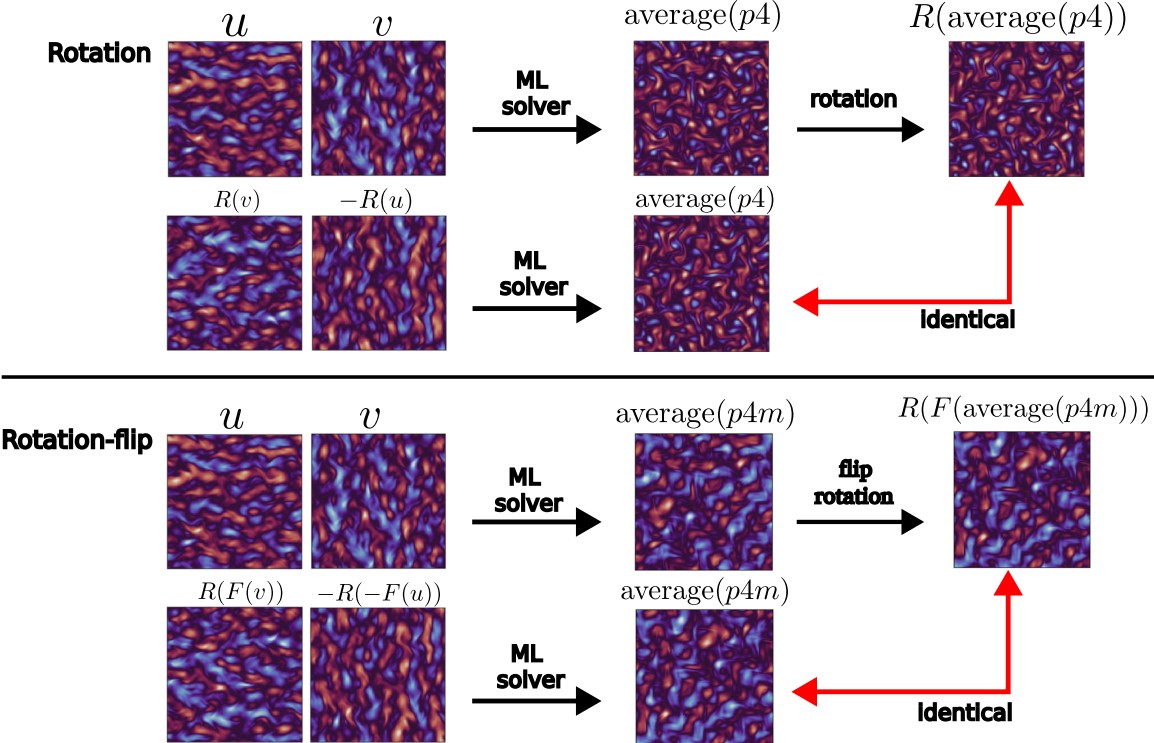

*Figure 12.* Group equivariance of INS input layers for p4 and p4m. The mathematical formulas of these input layers are given in equations 42-49.

## D. Group equivariant output layer on the C-grid staggering for INS

As shown in Fig. 1, the internal (hidden) layers of our equivariant PDE surrogates comprise a modern U-Net. In the case of a scalar field output defined at grid cell centers, direct utilization of the function `r2_act.trivial_repr` in escnn is permissible, for example, for $\zeta$ in SWEs. However, for vector fields with C-grid staggering, the vector field implementation in escnn, referred to as `r2_act.irrep(1)`, cannot be used because it is not on the C-grid and does not satisfy the symmetries of the discretized PDEs. Therefore, the output layers of p4 and p4m for the vector field are constructed on the C-grid in INS using the following formulations:

$$\mathbf{p4}: \ u_{i+0.5,j} = p_{i+1,j,0} - p_{i,j,1}; \quad v_{i,j+0.5} = p_{i,j+1,2} - p_{i,j,3} \tag{50}$$

$$\mathbf{p4m}: \begin{cases} u_{i+0.5,j} = p_{i+1,j,1} - p_{i,j,3} + p_{i+1,j,5} - p_{i,j,7} \\ v_{i,j+0.5} = p_{i,j+1,2} - p_{i,j,4} + p_{i,j+1,6} - p_{i,j,0} \end{cases} \tag{51}$$

For a regular representations $p$, the elements $p_{i,j,k}$ are defined as follows: The index $i$ and $j$ denote the spatial position, while $k$ indicates the group element index. $i + 0.5$ and $j + 0.5$ denote the position of outputs on the C-grids for $u$ and $v$, which are staggered by half a grid cell relative to $p$. These layers satisfy the group equivariance property while providing staggered outputs on the C-grid. Fig. (1) show how the output layer make up part of the surrogate architecture (blue box).

### D.1. Proofs of group equivariance for output layers with C-grid staggering

Equations 50-51 give the formulas defining the output layers for p4 and p4m. In this section, we will demonstrate that the equivariance relations are satisfied as intended.

#### D.1.1. VECTOR OUTPUT LAYER WITH P4 REGULAR REPRESENTATION INPUTS

Given inputs $u_{\text{in}}, vu_{\text{in}}$, the regular representation layer contains four channels indexed by the finite rotational component of p4 symmetry group: $p_{\cdot,\cdot,0}, p_{\cdot,\cdot,1}, p_{\cdot,\cdot,2}$, and $p_{\cdot,\cdot,3}$. When the input undergoes transformation to $R(v), -R(u)$, the four

regular representations are converted to $\hat{p}_{\cdot,\cdot,0}$, $\hat{p}_{\cdot,\cdot,1}$, $\hat{p}_{\cdot,\cdot,2}$ and $\hat{p}_{\cdot,\cdot,3}$ respectively. By the assumption of equivariance for all network components from the inputs to $p$, the following relations hold:

$$R(p_{\cdot,\cdot,0}) = \hat{p}_{\cdot,\cdot,3} \tag{52}$$
$$R(p_{\cdot,\cdot,1}) = \hat{p}_{\cdot,\cdot,0} \tag{53}$$
$$R(p_{\cdot,\cdot,2}) = \hat{p}_{\cdot,\cdot,1} \tag{54}$$
$$R(p_{\cdot,\cdot,3}) = \hat{p}_{\cdot,\cdot,2} \tag{55}$$

This equivariance relation is of the same form (staggered vector field mapped onto collocated regular representation) as the one investigated empirically in Fig. (2).

To prove the output layers defined by Eq. 50-51 are equivariant, we first consider a wider class of linear output layers, identifying constraints on their linear weights equivalent to equivariance, then choose our output layer among these. The wider class of output layers we consider (not all of which are equivariant), are defined by setting the outputs $u_{i+0.5,j}$, $v_{i,j+0.5}$ defined at grid cell interfaces to be equal to linear combinations to the regular representation values at the grid cell centers on either side of the interfaces. Thus, we have:

$$u_{i+0.5,j} = \sum_{k=0}^{3} c_k p_{i+1,j,k} - \sum_{k=0}^{3} d_k p_{i,j,k} \tag{56}$$

$$v_{i,j+0.5} = \sum_{k=0}^{3} e_k p_{i,j+1,k} - \sum_{k=0}^{3} f_k p_{i,j,k} \tag{57}$$

where $c_k$, $d_k$, $e_k$, and $f_k$ are freely chosen coefficients.

Rotate-transforming the inputs, we end up with a transformed regular representation $\hat{p}$ that provides input to the final output layer defined in Eq. 56-57, which produces final outputs $\hat{u}, \hat{v}$. We have equivariance if and only if:

$$(\hat{u}, \hat{v}) = (\mathcal{T}_R^{\text{vector}})^b [(u, v)] \tag{58}$$

where $0 \leq b < 4$ and $\mathcal{T}_R^{\text{vector}}$ is defined as in Eq. 5. By the definition of the rotation operator's action on regular representations and vector fields, we therefore arrive at the following constraints:

$$c_1 = d_2 = e_3 = f_4 \tag{59}$$
$$c_2 = d_3 = e_4 = f_1 \tag{60}$$
$$c_3 = d_4 = e_1 = f_2 \tag{61}$$
$$c_4 = d_1 = e_2 = f_3 \tag{62}$$

This the set of local linear readout layers (Eq. 56-57) that are p4 equivariant is 4-dimensional, and we could pick any of them or learn 4 coefficients. In this work, we opt for a simplicity, setting $c_1 = 1$ and $c_2 = c_3 = c_4 = 0$. Consequently, the equivariant vector output from the p4 regular representation can be expressed as:

$$u_{i+0.5,j} = p_{i+1,j,1} - p_{i,j,3} \tag{63}$$
$$v_{i,j+0.5} = p_{i,j+1,2} - p_{i,j,4} \tag{64}$$

It is important to note that this is merely one particular form of output layer for vector fields on the C-grid staggering. Other forms of the output layer are attainable by selecting different values for $c_i$, $d_i$, $e_i$, and $f_i$, which can also be learned.

### D.1.2. VECTOR OUTPUT LAYER WITH P4M REGULAR REPRESENTATION INPUTS

The proof for this case proceeds similarly to the p4 case. We begin by again considering a class of linear output layers mapping from the collocated regular representation to the staggered velocity fields. But whereas in Eq. 56-57 $c, d, e, f$ each have four coefficients, for p4m they have eight each according to the group dimension of the regular representation:

$$u_{i+0.5,j} = \sum_{k=0}^{7} c_k p_{i+1,j,k} - \sum_{k=0}^{7} d_k p_{i,j,k} \tag{65}$$

$$v_{i,j+0.5} = \sum_{k=0}^{7} e_k p_{i,j+1,k} - \sum_{k=0}^{7} f_k p_{i,j,k} \tag{66}$$

Here we have equivariance if and only if:

$$(\hat{u}, \hat{v}) = (\mathcal{T}_R^{\text{vector}})^b \circ (\mathcal{T}_F^{\text{vector}})^a [(u, v)] \tag{67}$$

where $0 \leq a < 2, 0 \leq b < 4$ and $\mathcal{T}_F^{\text{vector}}$ is the flipping operator for vector fields. Equating elements of the transformed outputs as required for each of the 8 possible transformations, we arrive at the constraints:

$$c_1 = d_2 = e_3 = f_4 = c_5 = d_6 = e_7 = f_0 \tag{68}$$
$$c_2 = d_3 = e_4 = f_1 = c_6 = d_7 = e_0 = f_5 \tag{69}$$
$$c_3 = d_4 = e_1 = f_2 = c_7 = d_0 = e_5 = f_6 \tag{70}$$
$$c_4 = d_1 = e_2 = f_3 = c_0 = d_5 = e_6 = f_7 \tag{71}$$

As before, any choice of $c$ will satisfy equivariance. We again take the simple case of $c_1 = 1$ and $c_2 = c_3 = c_4 = 0$. Thus, the vector output layers on C-grids for $u$ and $v$ from the p4m regular representation are written as follows:

$$u_{i+0.5,j} = p_{i+1,j,1} - p_{i,j,3} + p_{i+1,j,5} - p_{i,j,7} \tag{72}$$
$$v_{i,j+0.5} = p_{i,j+1,2} - p_{i,j,4} + p_{i,j+1,6} - p_{i,j,0}. \tag{73}$$

Note that this is only one possible form for the output layer.

## E. Group equivariance of PDE surrogate architectures

As illustrated in Fig. 13, the group equivariance of a neural network requires equivariance for the input layer, internal layers and the output layer. Fig. 13a shows a network incorporating specialized input/output layers for velocity fields on a staggered C-grid. Here the network exhibits a precise equivariance relationship that holds up to numerical precision. Fig. 13b show the same network, but with a standard equivariant layer designed to operate on vector fields without staggering. Here the equivariance relation no longer holds.

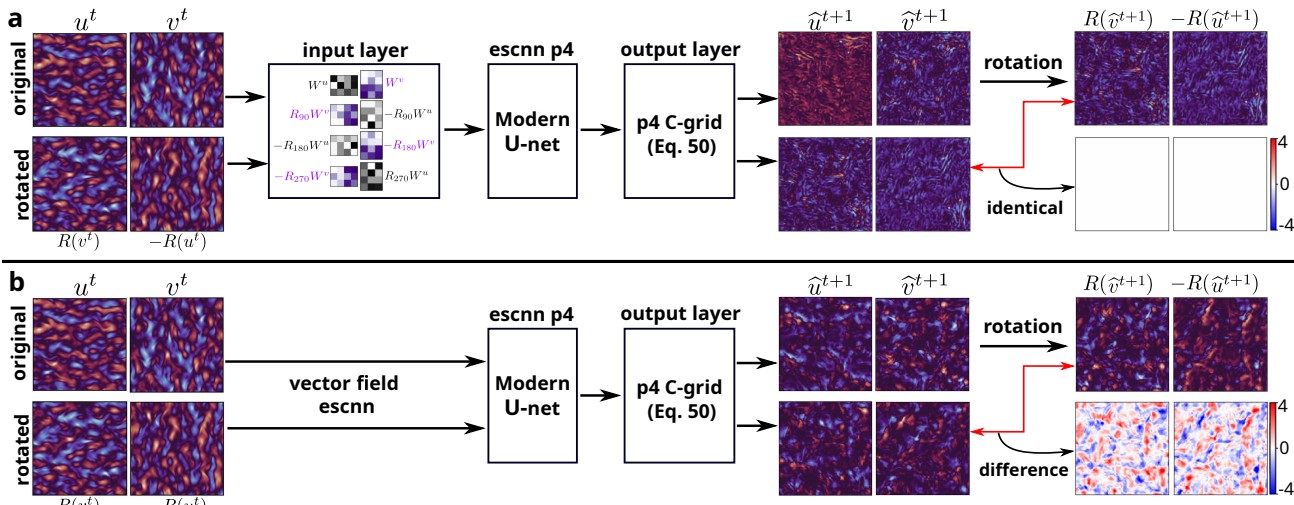

*Figure 13.* Equivariance of neural PDE surrogates with (a) and without (b) specialized input layers for velocity fields on staggered C-grids. Both networks used the same modern u-net architecture, but for (b) a standard equivariant vector field input from *escnn* was used, which assumes a collocated grid. Both networks were tested for equivariance in a randomly initialized state, without any training. The same input fields were used as input both before (upper rows) and after transforming the vector fields using a 90 degree rotation (bottom rows). Both networks use an output layer designed for velocity fields on staggered C-grid. For the network in (a), rotation-transformation of the output fields produced by original input fields matched the output fields arising from rotation-transformed inputs up to machine precision. For the network in (b), the equivariance relation did not hold.

## F. Imposing physical constraints on neural PDE surrogates

In the shallow water system, mass is a conserved variable. To ensure the preservation of this variable during training and inference, the mean of the update to the mass field is subtracted before the update is applied:

$$\zeta^{t+1} = \zeta^t + d\zeta - \text{mean}(d\zeta) \tag{74}$$

To conserve momentum without conserving mass in the INS system, perform a similar global mean correction.

$$u^{t+1} = u^t + du - \text{mean}(du); \quad v^{t+1} = v^t + dv - \text{mean}(dv) \tag{75}$$

To also impose mass conservation (equivalent to the divergence-free property of the velocity field), We learning a scalar potential $a$ and define the velocity field as the curl of this scalar field: $[\hat{u}^{t+1}, \hat{v}^{t+1}] = \nabla \times [0, 0, a]$ as in Wandel et al., 2021.

$$u^{t+1} = u^t - \frac{\partial a}{\partial y}; \quad v^{t+1} = v^t + \frac{\partial a}{\partial x}. \tag{76}$$

These physical constraint layers are added after the equivariant output layers as needed. An example implementation for INS is shown in the blue box in Fig. (1).

As an alternative approach, we might have also learned fluxes at the C-grid interfaces for conserved quantities at the cell centers, or fluxes at the vertices for conserved quantities at the interfaces. This approach would be analogous to a finite volume solver. A key advantage of this approach is that it is computed locally (McGreivy & Hakim, 2023), which avoids unphysical action at large distances and facilitates easier generalization of the domain size after training. We leave this avenue of exploration for future work, with the expectation that further improvements in accuracy could be achieved.

## G. Simulation parameters

In this section, we list simulation parameters used to solve the shallow water equations and the incompressible Navier-Stokes equations in the numerical solvers. We also show additional settings used to generate training data. Table 6 shows the parameters for solving SWEs. Table 7 shows the parameters used to solve the INS and to generate the training data.

*Table 6.* Simulation parameters for SWEs

| Parameters | Explanation | Value |
|---|---|---|
| $L \times L$ | simulation domain | $1000 \times 1000 \, (Km)$ |
| $d$ | undisturbed water depth | $100 \, (m)$ |
| $C_D$ | bottom drag coefficient | $1.0e - 3$ |
| $g$ | acceleration due to gravity | $9.81 \, (m/s^2)$ |
| $\Delta x$ | space step | $10 \, (Km)$ |
| $\Delta t$ | time step | $300 \, (s)$ |
| $w_{\mathbf{imp}}$ | implicit weighting | $0.5$ |

*Table 7.* Simulation- and data generation parameters for INS

| Parameters | Explanation | Value |
|---|---|---|
| $L \times L$ | simulation domain | $2\pi \times 2\pi$ |
| $\rho$ | density | $1$ |
| $\mu$ | viscosity | $1e - 3$ |
| $T$ | simulation time | $224.34 \, s$ |
| $\Delta t_{\text{solver}}$ | the time step of numerical solver | $0.00436 \, s$ |
| $M \times M$ | the grids of numerical solver | $576 \times 576$ |
| $\Delta x_{\text{solver}}$ | the space step of numerical solver | $0.0109$ |
| $\Delta t_{\text{ml}}$ | the time step of ML model | $0.8375$ |
| $m \times m$ | the grids of ML model | $48 \times 48$ |
| $\Delta x_{\text{ml}}$ | the space step of ML solver | $0.1308$ |

## H. Architectural details of (group equivariant) modern U-Net

We modified the Modern U-Net (Gupta & Brandstetter, 2023) to create a symmetry- and physics-constrained version. Table 8 describes network hyperparameters, and how these were adjusted depending on the chosen symmetry group in order to match total parameter counts.

*Table 8.* Detailed architectural settings for (group equivariant) modern U-Net. In this table, we compare the hyperparameters of a modern U-Net with those of p4- and p4m-equivariant versions.

| Hyperparameter | Modern U-net | Modern U-net p4 | Modern U-net p4m |
|---|---|---|---|
| Number of time steps in the input (time_history (int)) | 1 | 1 | 1 |
| Number of time steps in the output (time_future (int)) | 1 | 1 | 1 |
| Base hidden layer channel count (before multiplier) | 4 | 2 | 2 |
| Activation function | Gelu | Gelu | Gelu |
| Normalization | True | True | True |
| List of channel multipliers for each resolution | (1,2,4) | (1,3,3) | (1,4,1) |
| Self-attention used at bottleneck | False | False | False |
| Number of residual blocks at each resolution | 1 | 1 | 1 |
| Kernel size | (3, 3) | (3, 3) | (3, 3) |
| Padding | (1, 1) | (1, 1) | (1, 1) |
| Padding mode | Circular | Circular | Circular |
| Convolution type for internal layers | Conv2d | rot2dOnR2(N=4) | flipRot2dOnR2(N=4) |

## I. A hybrid method for the inference of shallow water system

Figure 14 shows a hybrid method used to predict the solution of a shallow water system. In our neural PDE surrogate, we have only one output $\zeta$ and we have three inputs $u$, $v$ and $\zeta$. An additional physics-based solution step is required to compute $u^t$ and $v^t$ from the $\zeta^t$ produced by the surrogate. These additional calculations are performed only for autoregressive rollouts with trained networks, not during training. Following Backhaus, 1983, we first calculate an interim solution $u^*$, $v^*$ using an explicit time step:

$$u^* = u^n - \Delta t c_D \frac{1}{h} u^n |u^n| - \Delta t g (1 - w_{\mathbf{imp}}) \frac{\partial \zeta^n}{\partial x} + \Delta t a_h \nabla^2 u^n \tag{77}$$

$$v^* = v^n - \Delta t c_D \frac{1}{h} v^n |v^n| - \Delta t g (1 - w_{\mathbf{imp}}) \frac{\partial \zeta^n}{\partial y} + \Delta t a_h \nabla^2 v^n \tag{78}$$

here the superscript $n$ indicates the time step. Next, the interim velocities are combined with the surrogate's estimate of $\zeta$ to produce a semi-implicit solution for the velocities:

$$u^{n+1} = u^* - \Delta t g w_{\mathbf{imp}} \frac{\partial \zeta^{n+1}}{\partial x} \tag{79}$$

$$v^{n+1} = v^* - \Delta t g w_{\mathbf{imp}} \frac{\partial \zeta^{n+1}}{\partial y} \tag{80}$$

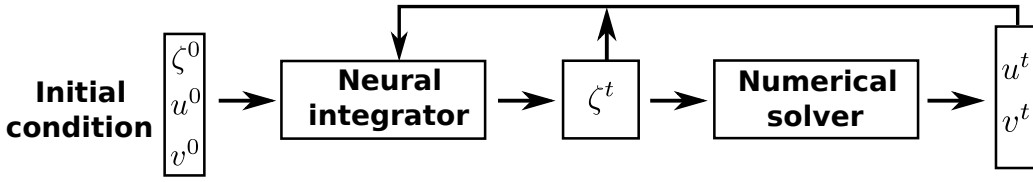

*Figure 14.* The figure describes the configuration of the hybrid method for the inference of shallow water systems from a trained model. The fluid surface elevation $\zeta^t$ is predicted by the neural network. The calculation of $u^t$ and $v^t$ is achieved through the application of the following Equations (79-80) with the known value of $\zeta^t$.

## J. Additional results

### J.1. SWE rollouts

Fig. 15 shows predicted surface heights $\zeta$ for the shallow water system with additional time steps and models. The primary text presents two models, p1/∅ and p4m/M, which are evaluated against the reference at six time steps, as illustrated in Figure 3-a. Figure 15 extends this by presenting the reference, p1/∅, p1/M, p1/∅ + $\epsilon$, p4/∅, p4/M, p4m/∅, p4m/M, p4m/M+$\epsilon$, FNO/∅, and drnet/∅ with the reference at nine time steps.

These rollouts illustrate only one of thirty distinct initial conditions. In this particular instance, the p4m/M model is the most stable and accurate model in comparison to all other methods. As previously mentioned, the main text also displayed statistical scores calculated over twenty distinct initial conditions, for example in Figure 15c-g.

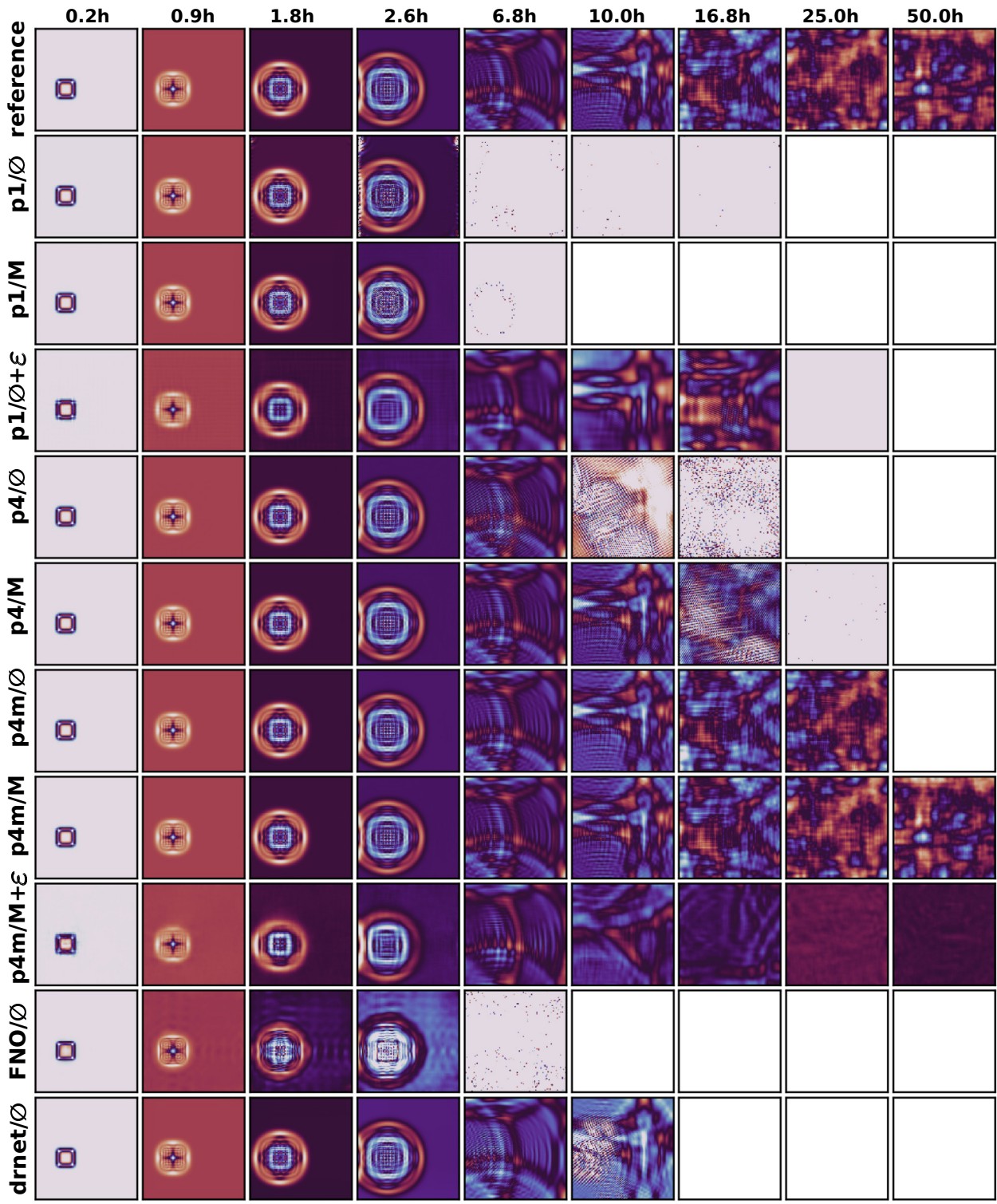

*Figure 15.* Rollout predictions for the fluid surface elevation $\zeta$ for all surrogates on the closed boundary shallow water system at various forecast horizons. The predictions of ten trained models are compared to the reference. The results obtained from this analysis clearly demonstrate that the p4m/M method exhibits superior performance in terms of stability and accuracy when compared to all other methods. This figure is an expanded version of Fig. 3a in the main text.

### J.2. Training with input noise for SWE

We compare our unconstrained and physics/symmetry-constrained SWE surrogates, with their noisy variants p1/∅, p1/∅ + $\epsilon$ (Fig. 16). We added Gaussian noise drawn from $\mathcal{N}(\mu = 0, \sigma = 0.0001)$ during the training process. Training with input noise yielded moderately low error for long very rollouts, but yielded lower accuracy compared to the noise-free model for shorter rollouts, especially when constraints were used. Predictions from the noisy model demonstrate reduced accuracy, even at the early stages of the rollout.

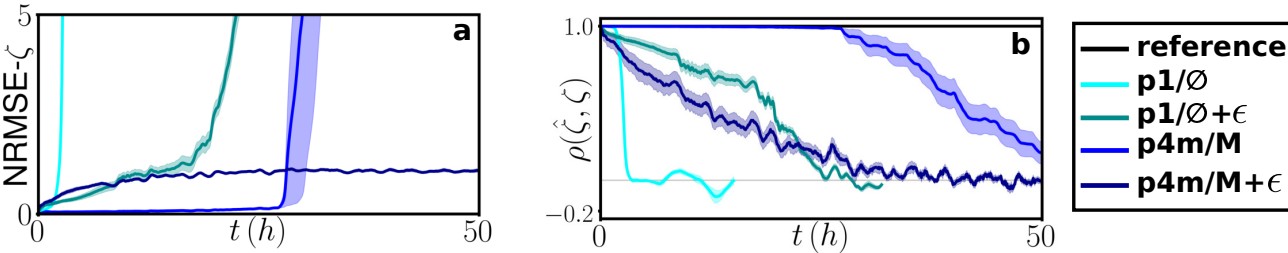

*Figure 16.* Comparison of reference simulations and neural surrogates trained with input noise, denoted p1/∅ + $\varepsilon$ and p4m/M+$\varepsilon$, as well as their their noise-free counterparts p1/∅ and p4m/M, based on the metrics NRMSE-$\zeta$ and $\rho(\hat{\zeta}, \zeta)$. The NRMSE-$\zeta$ metric reveals that p4m/M+$\epsilon$ exhibits a comparatively diminished error over an extended time span in comparison to alternative methodologies. p4m/M delivers a more precise prediction for the first 25 hours.

### J.3. Time evolution of conserved quantities for SWE

Fig. 17 shows the mass, momentum, and total energy for the closed boundary shallow water system for various surrogates and the reference solution over the course of 50 simulated hours 17. The mass and total energy remain constant, while momentum oscillates slightly (black curves). The p4m/M model gives the closets match to the reference solution.

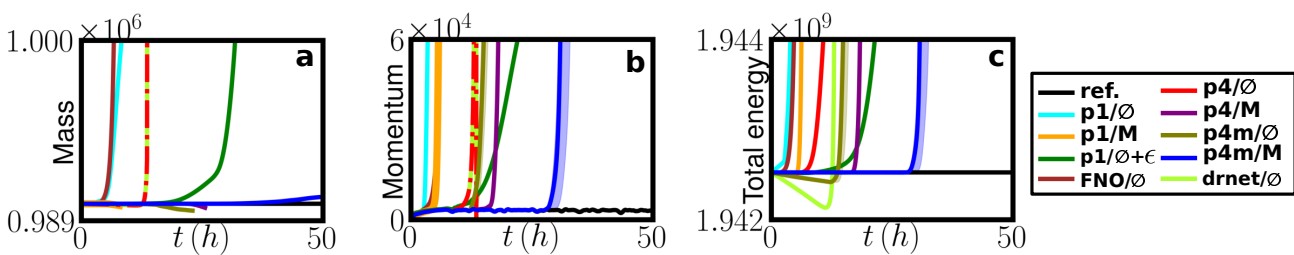

*Figure 17.* Mass, momentum and total energy for the shallow water equations for all surrogates tested over the course of 50 simulated hours. The p4m/M model, which has been identified as the most effective model, exhibits superior performance in reproducing the mass, momentum and total energy of the solver.

### J.4. Decaying turbulence rollouts

Fig. (18-19), show rollout of velocities $u$ and $v$ for additional time points and surrogates, extending Fig. 4 from the main text. We compared the predictions of velocity $u$ and $v$ from thirteen surrogates with the reference solution: p1/∅, p1/$\rho\vec{u}$, p1/M+$\rho\vec{u}$, p1/∅ + $\epsilon$, p4/∅, p4/$\rho\vec{u}$, p4/M+$\rho\vec{u}$, p4m/∅, p4m/$\rho\vec{u}$, p4m/M+$\rho\vec{u}$, p4m/M+$\rho\vec{u}$ + $\epsilon$, FNO/∅ and drnet/∅. These results show just one of 30 initial conditions.

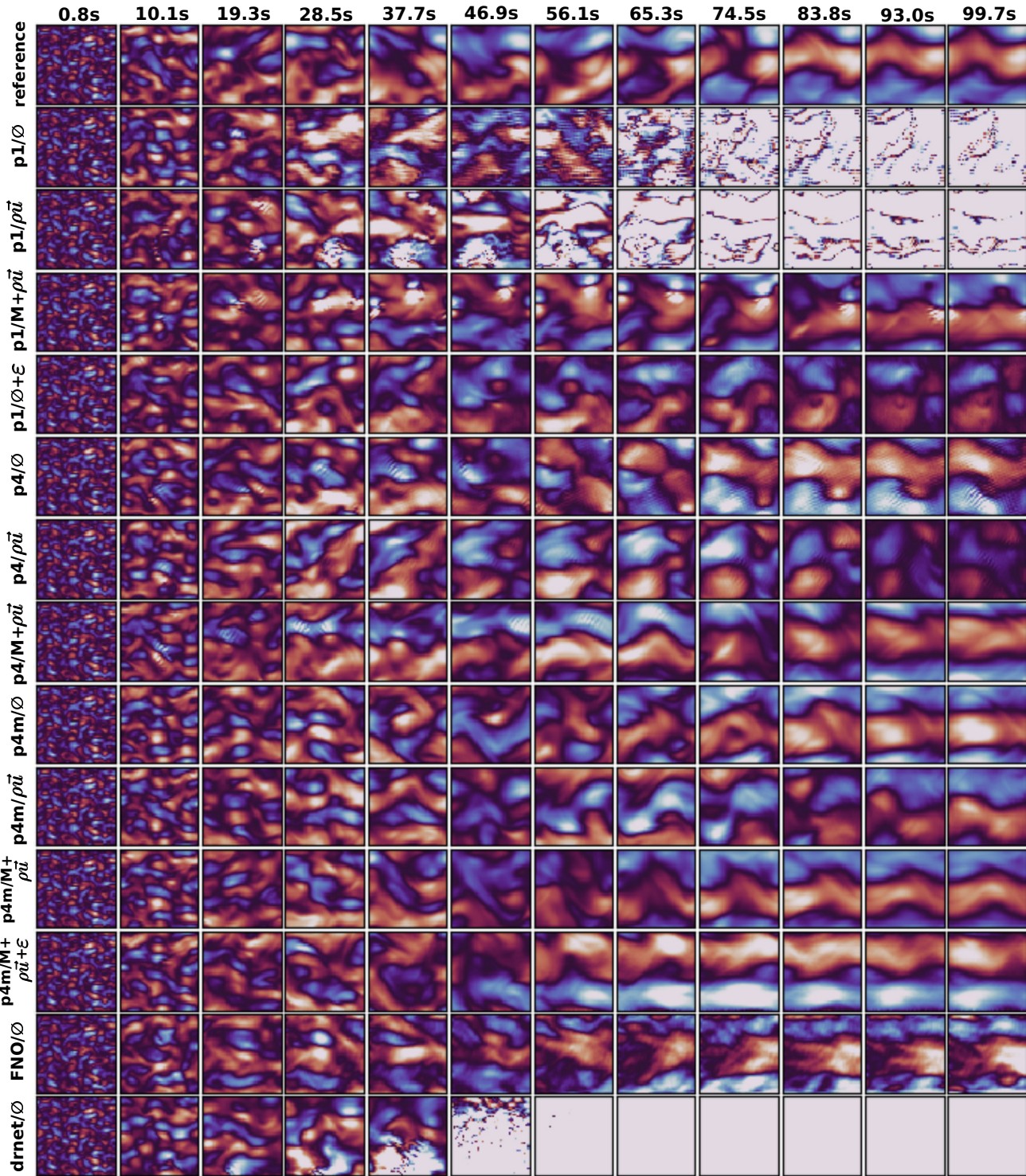

*Figure 18.* Rollout predictions for $u$ from thirteen models with approximately 0.1M parameters each for decaying turbulence at varying time steps. The top row shows the reference simulation. This plot extends Fig. 4 from the main text.

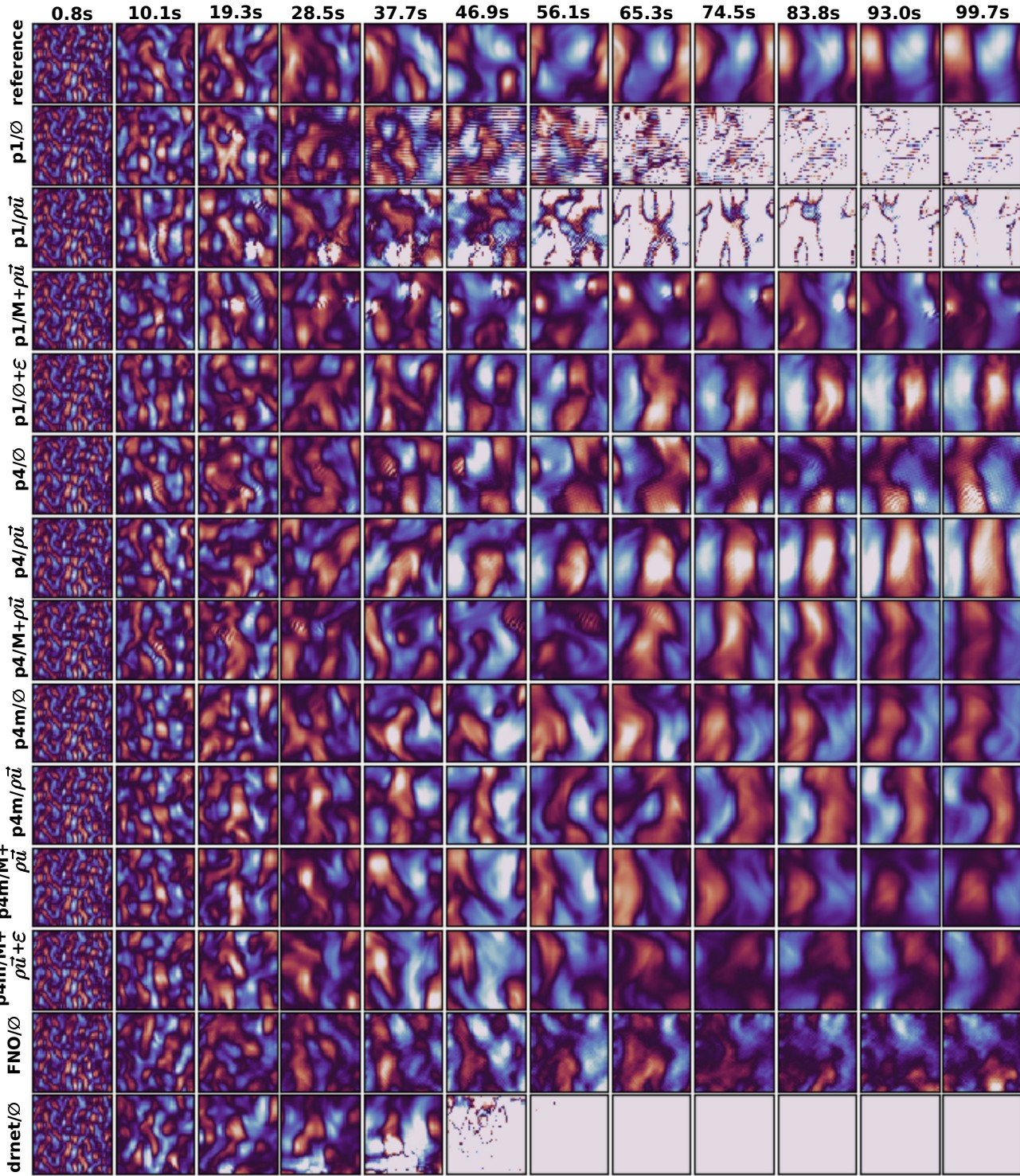

*Figure 19.* Rollout predictions for $v$ from thirteen models with approximately 0.1M parameters each for decaying turbulence at varying time steps. The top row shows the reference simulation.

## J.5. Momentum in surrogate rollouts for decaying turbulence

Figure. 20 shows reference and predicted momentum for various surrogates. The non-physically-constrained models (p1/∅, p4/∅, and p4m/∅) exhibit a substantial increase in momentum compared to the reference.

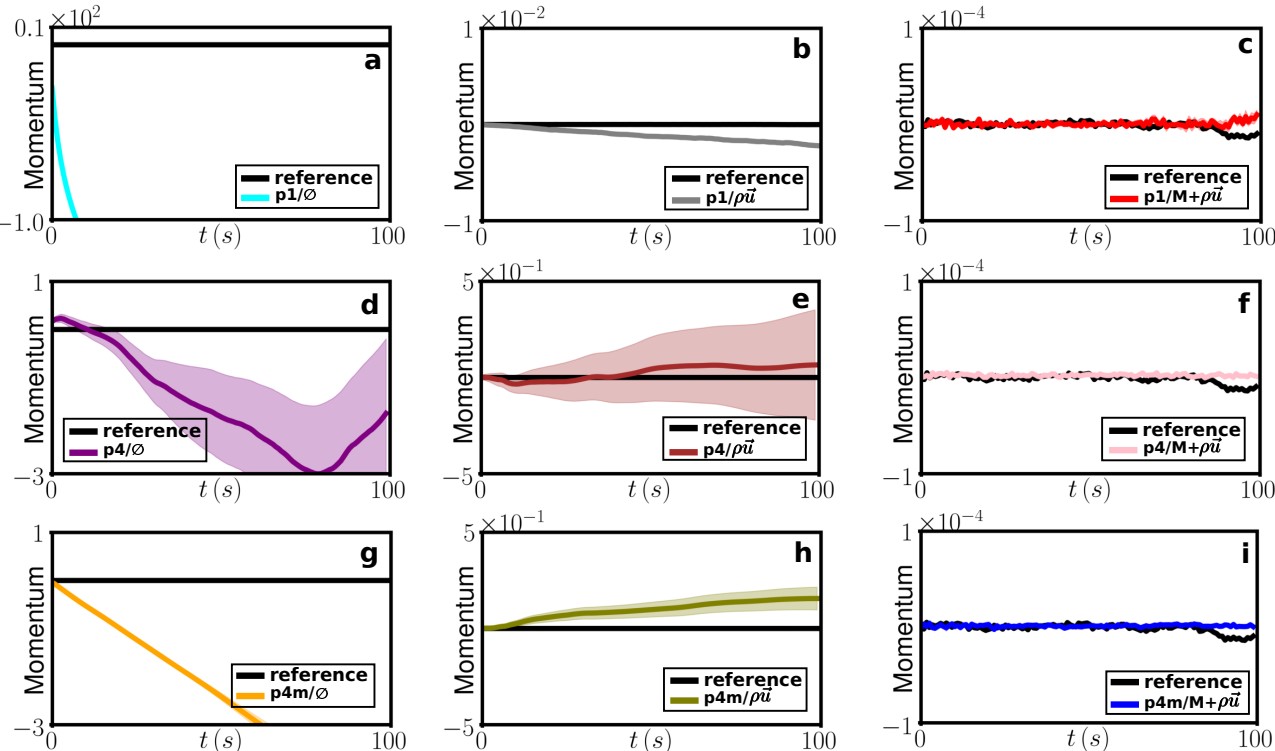

*Figure 20.* Momentum over time for all surrogates on the decaying incompressible turbulence task, compared to momentum of reference solution (black).

## J.6. Velocity and Energy Spectra for Decaying Turbulence

Figure (21) shows the velocity and energy spectra ($u(k)k^5$ and $E(k)k^5$) at additional time steps, expanding on the two time steps previously shown in Figure 4-d,e of the main text. Our analysis demonstrates that the model p4m/M+$\rho\vec{u}$ (blue curve) consistently provides the closest match to the reference across all time steps for both the velocity and energy spectra.

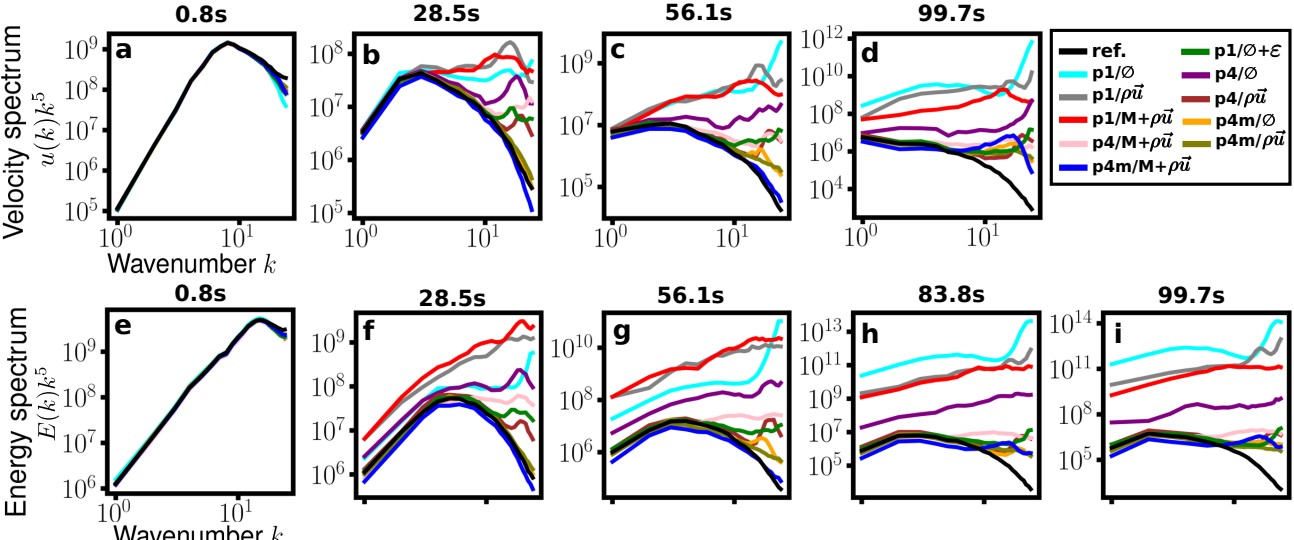

*Figure 21.* Velocity and energy spectra at additional time points. Results are consistent with the those presented in the main text, specifically Figures (4-d,e). p4m/M+$\rho\vec{u}$ (blue curves) most closely matches the reference spectra (black).

### J.7. Training with input noise for decaying turbulence

We investigated the effect of input noise during the training process for the decaying turbulence task (Fig. 22). We trained with input noise drawn from $\mathcal{N}(\mu = 0, \sigma = 0.001)$. Training with input noise did not produce a clear increase in accuracy for unconstrained surrogates.

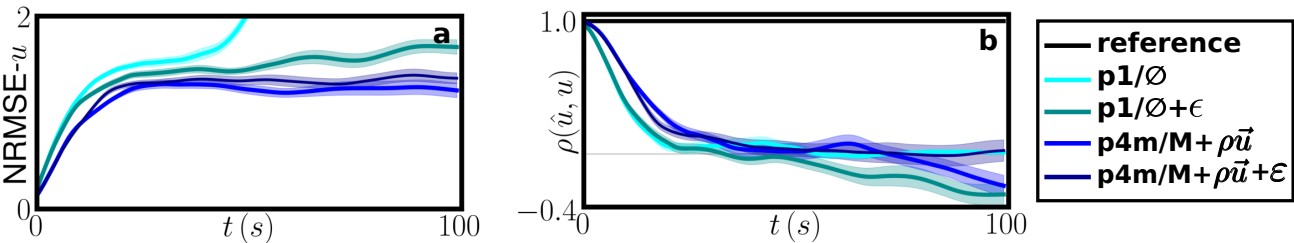

*Figure 22.* Comparison of surrogates trained with noise vs. clean data: p1/$\varnothing + \varepsilon$, p4m/M+$\rho\vec{u} + \varepsilon$, p1/$\varnothing$ and p4m/M+$\rho\vec{u}$. We find that training with noise did not bring about clear improvements to accuracy, though unconstrained models were slightly more accurate for long rollouts.

### J.8. Dilated ResNet as surrogate for decaying turbulence

In the main text, we used the robust Modern U-net architecture for the study of INS. In this section, an alternative architecture, DilatedRes Net Stachenfeld et al., 2021; Kohl et al., 2024, is modified to incorporate equivariance with respect to p4 and p4m. The input layer, output layer, and physical constraints of these models are consistent with those outlined in the main text. In this section, we focus on investigating nine Dilated ResNet models: p1/$\varnothing$, p1/$\rho\vec{u}$, p1/M+$\rho\vec{u}$, p4/$\varnothing$, p4/$\rho\vec{u}$, p4/M+$\rho\vec{u}$, p4m/$\varnothing$, p4m/$\rho\vec{u}$, p4m/M+$\rho\vec{u}$.

Fig. (23-24) present a single instance of velocities $u$ and $v$ predicted by nine Dilated ResNet models, as well as the reference simulation. The network size of these models is approximately 0.1M, which is comparable to the modern U-Net in Figure 4-a. The p4m/M+$\rho\vec{u}$ with Dilated ResNet exhibits the best rollouts for both $u$ and $v$ when compared to the reference, as illustrated in Figures (23-24).

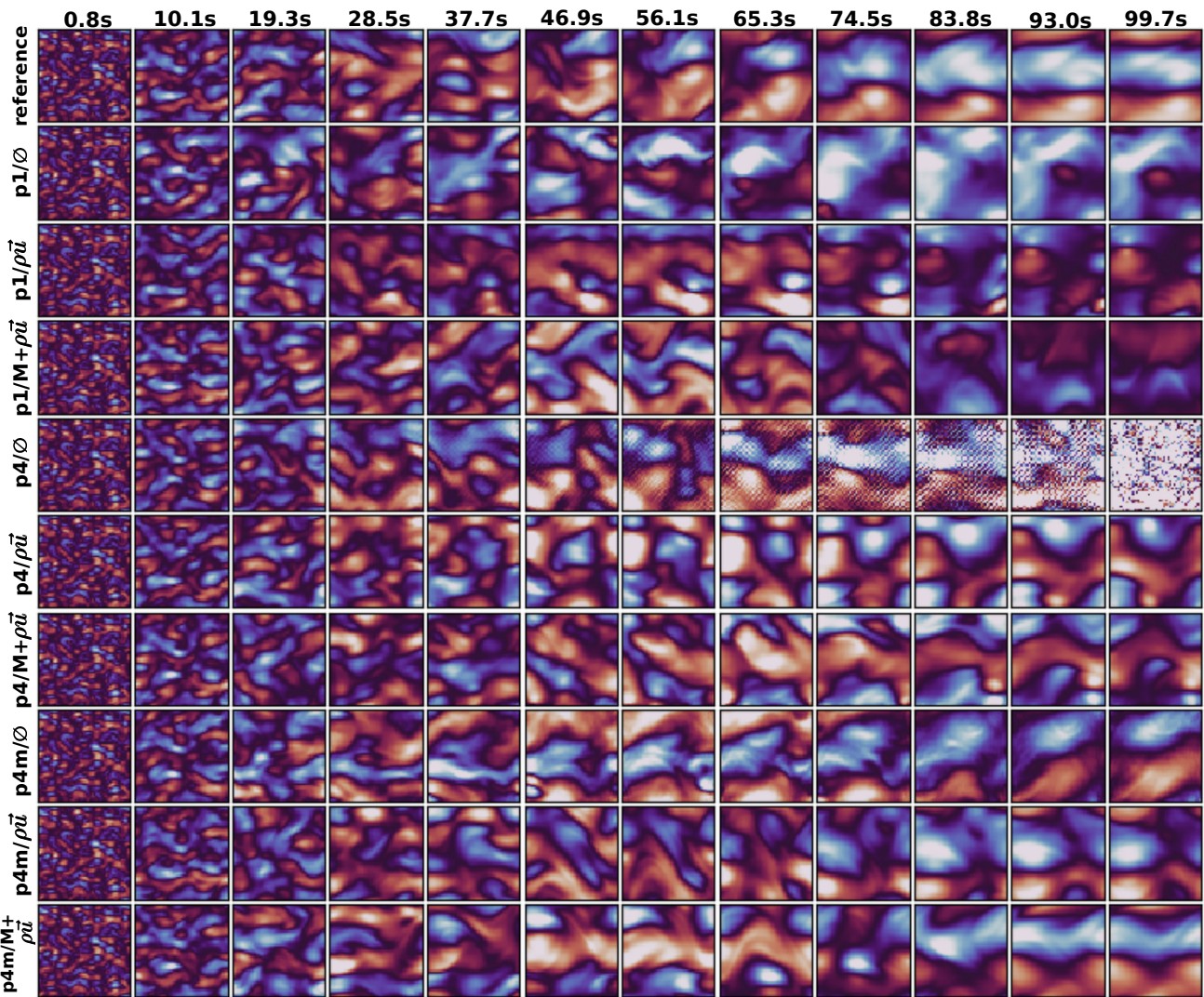

*Figure 23.* Rollouts of $u$ by nine Dilated ResNet surrogates for decaying turbulence. The network size these surrogates was approximately 0.1M parameters. In this particular instance, the p4m/M+$\rho\vec{u}$ model is the most similar to the reference solution.

*Figure 24.* Rollouts of $u$ by nine Dilated ResNet surrogates for decaying turbulence, same data as Fig. 23.

Over 30 random initial conditions, we computed accuracy metrics for these drnet surrogates (Fig. 25). The worst dilated ResNet model was p1/$\varnothing$, and the best was p4m/M+$\rho\vec{u}$.

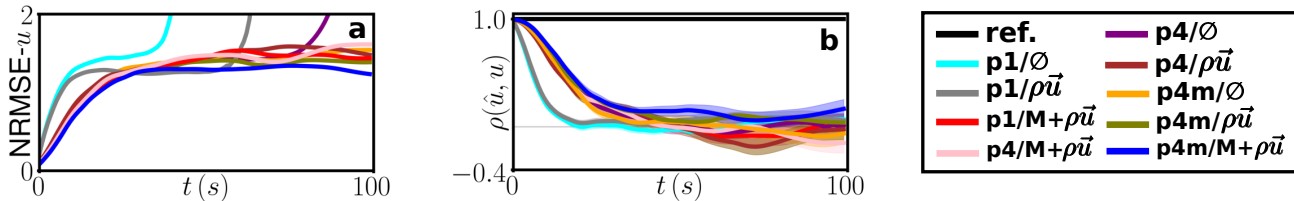

*Figure 25.* NRMSE-$u$ and $\rho(\hat{u}, u)$ for the Dilated ResNet surrogates on the decaying turbulence task. Results are derived from 30 random initial conditions.

We also computed velocity and energy spectra ($u(k)k^5$ and $E(k)k^5$) for the Dilated ResNet models of decaying turbulence. Doubly constrained surrogates reproduced these spectra most faithfully (Fig. 26.

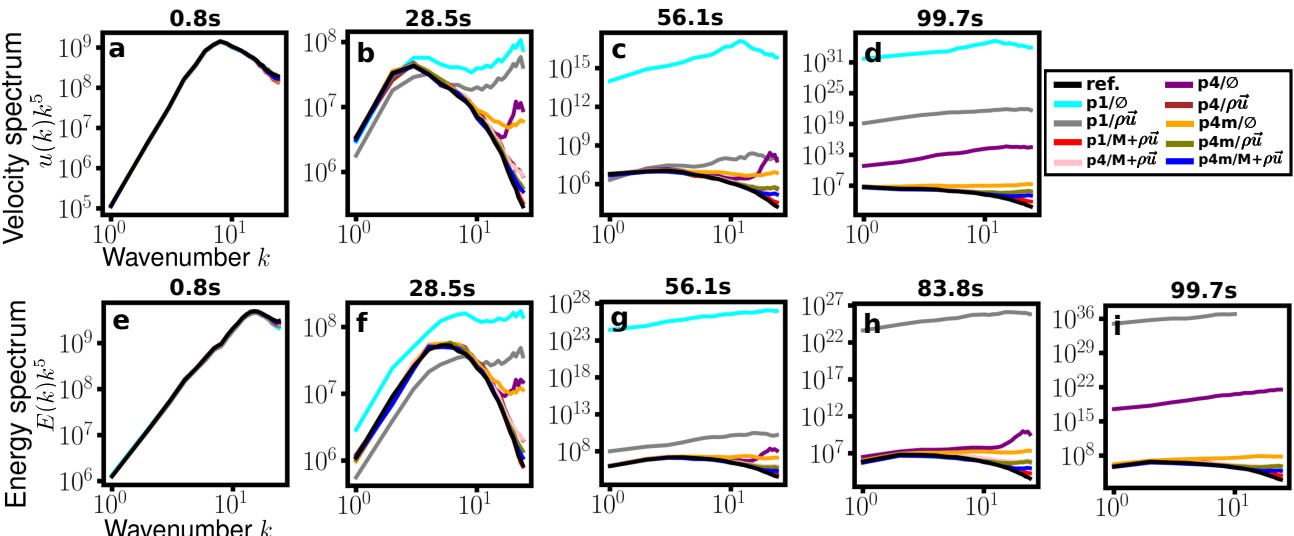

*Figure 26.* Velocity and energy spectra ($u(k)k^5$ and $E(k)k^5$) for the Dilated ResNet models of decaying turbulence.

### J.9. Pushforward Training

Here we analyze surrogates trained with the pushforward trick (Brandstetter et al., 2022c) in greater detail. We consider the modern U-Net base architecture and its constrained variants on the decaying turbulence task. Pushforward training employs an auto-regressive rollout, using model outputs as inputs for the next time step time step, but does not propagate gradients backwards in time. Following Brandstetter et al., 2022c, we roll out for two time steps.

With PF training, the maximally constrained model p4m/M+$\rho\vec{u}$+PF exhibited significantly reduced NRMSE compared to less constrained models, while the improvement was less clear for correlation.

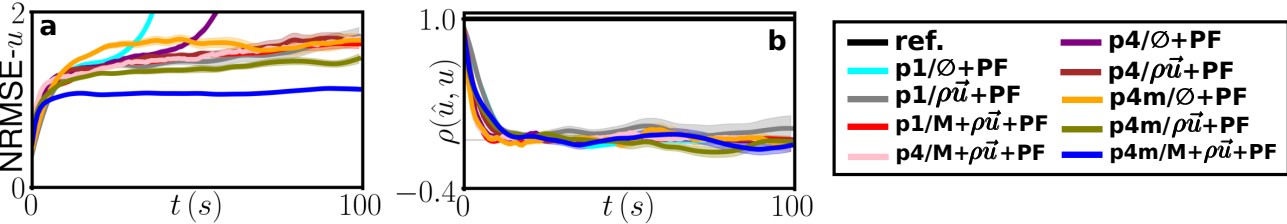

*Figure 27.* NRMSE-$u$ and $\rho(\hat{u}, u)$ for pushforward training of PDE surrogates.

### J.10. Data augmentation

Data augmentation is a regularization technique that uses transformed copies of data points to train a neural network, to reduce overfitting and improve generalization. It has been employed to train PDE surrogates (Brandstetter et al., 2022b; Fanaskov et al., 2023). We augment with random rotations, as well as flipping with probability 0.5. The transformation is applied to both inputs and outputs for each training data point, with transformations applied to vector fields as previously described.

As expected, data augmentation had no effect on fully equivariant surrogates, but did bring about some improvement in non equivariant models (Fig. 28).

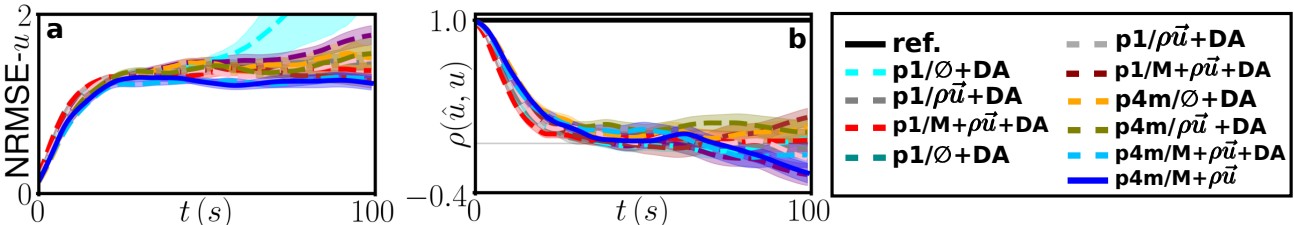

*Figure 28.* Performance of data-augmented models.

## J.11. Generalization tasks

We tested surrogates on additional ICs that were not present in their training datasets, for the SWE and INS cases.

### J.11.1. GENERALIZATION FOR SWES

We evaluated generalization capability for SWE surrogates in 3 experiments with 20 ICs each. First, we used a single rectangle with a size of 0.1 m as the IC, with its location chosen at random (Fig. 29). Second, we used the sum of two rectangles, each measuring 0.1 m, with their locations selected at random (Fig. 30). Finally, we used two overlapping rectangular elevations (summed over the region of overlap, Fig.31), which proved the most challenging configuration. The p4m/M surrogate proved superior in each case.

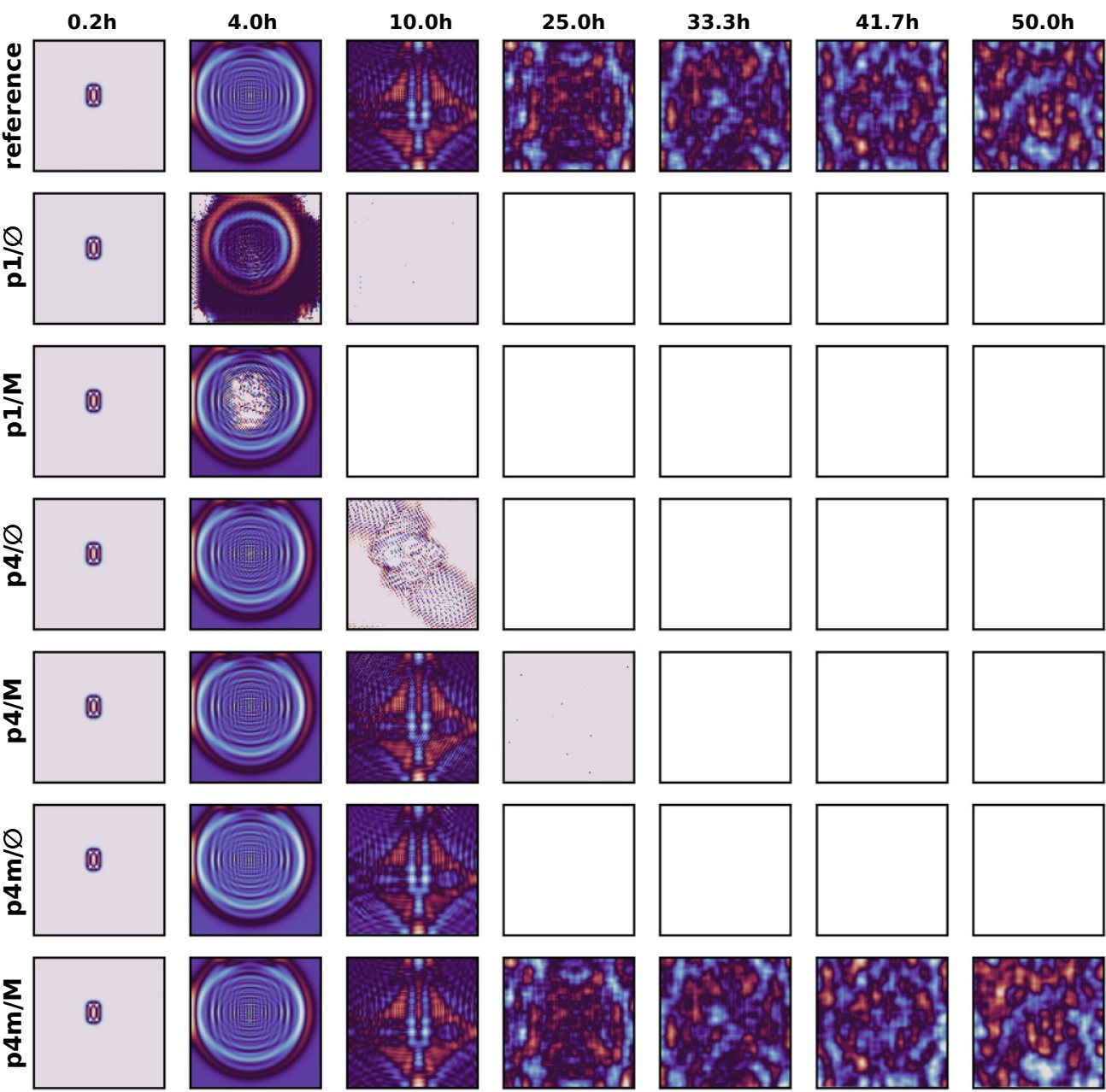

*Figure 29.* Rollouts demonstrating generalization for the SWEs, generated by all surrogates from a single rectangular-shaped elevation IC, are shown at various time intervals.

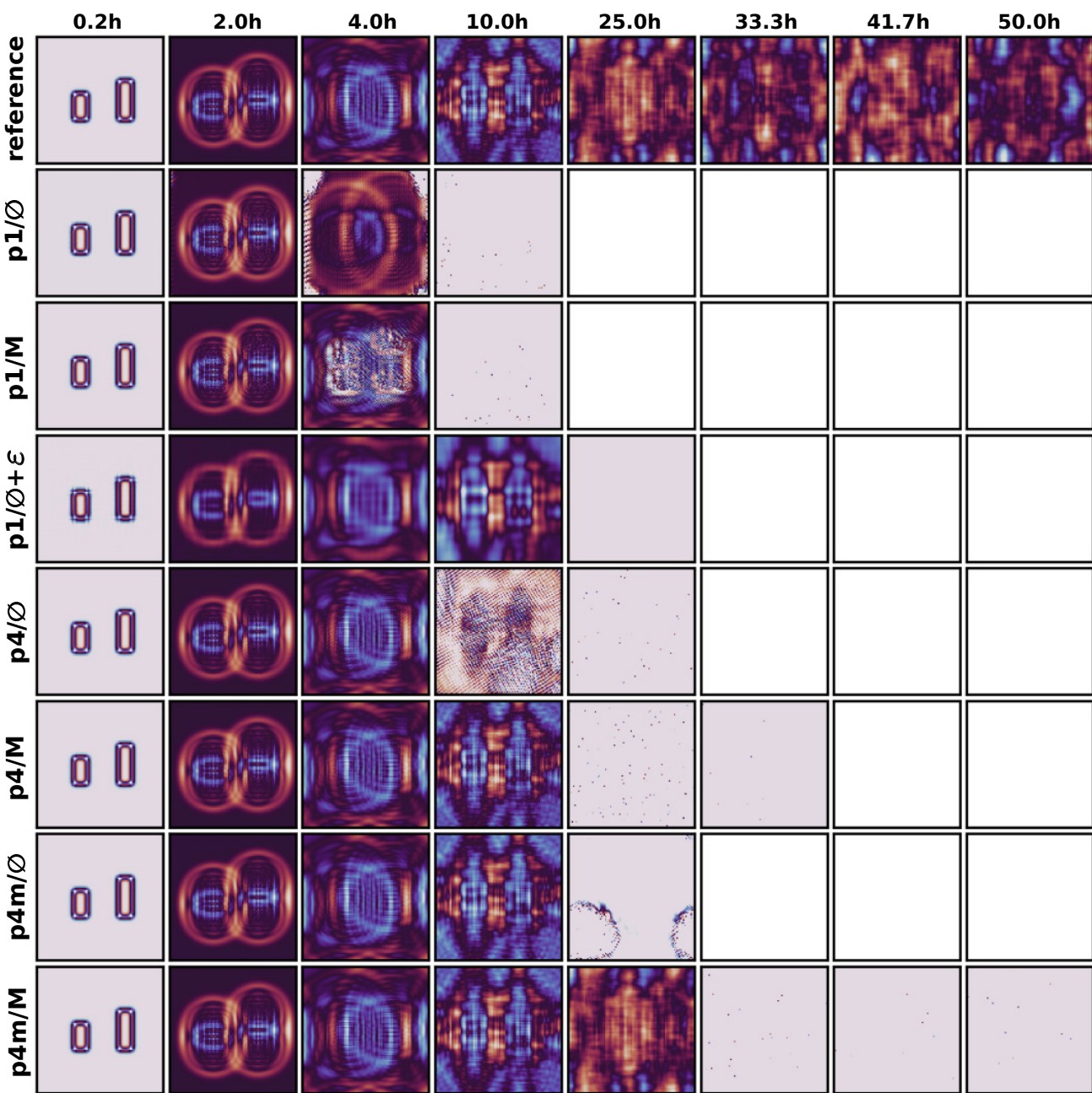

*Figure 30.* Rollouts demonstrating generalization for the SWEs, generated by all surrogates from a single IC consisting of two rectangular elevations, are shown at various time intervals.

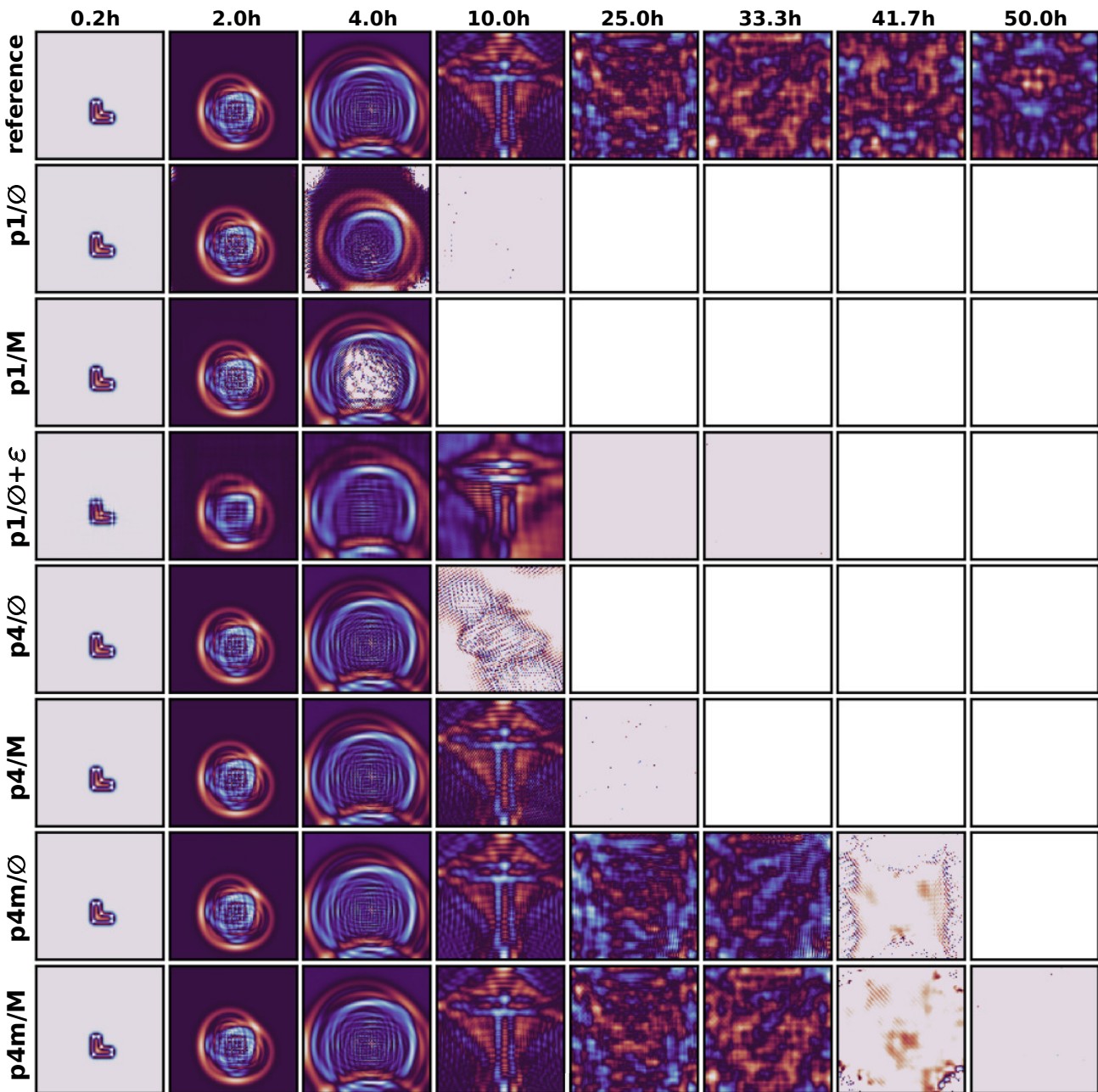

*Figure 31.* Rollouts demonstrating generalization for the SWEs, generated by all surrogates from a single IC consisting of two overlapping (summed) rectangular elevations, are shown at various time intervals.

### J.11.2. GENERALIZATION FOR DECAYING TURBULENCE

We evaluated generalization capability for INS surrogates, using 10 ICs with peak wavenumber 8. Figs. 32-33 show examples of the reference solution and surrogate rollouts for this data.

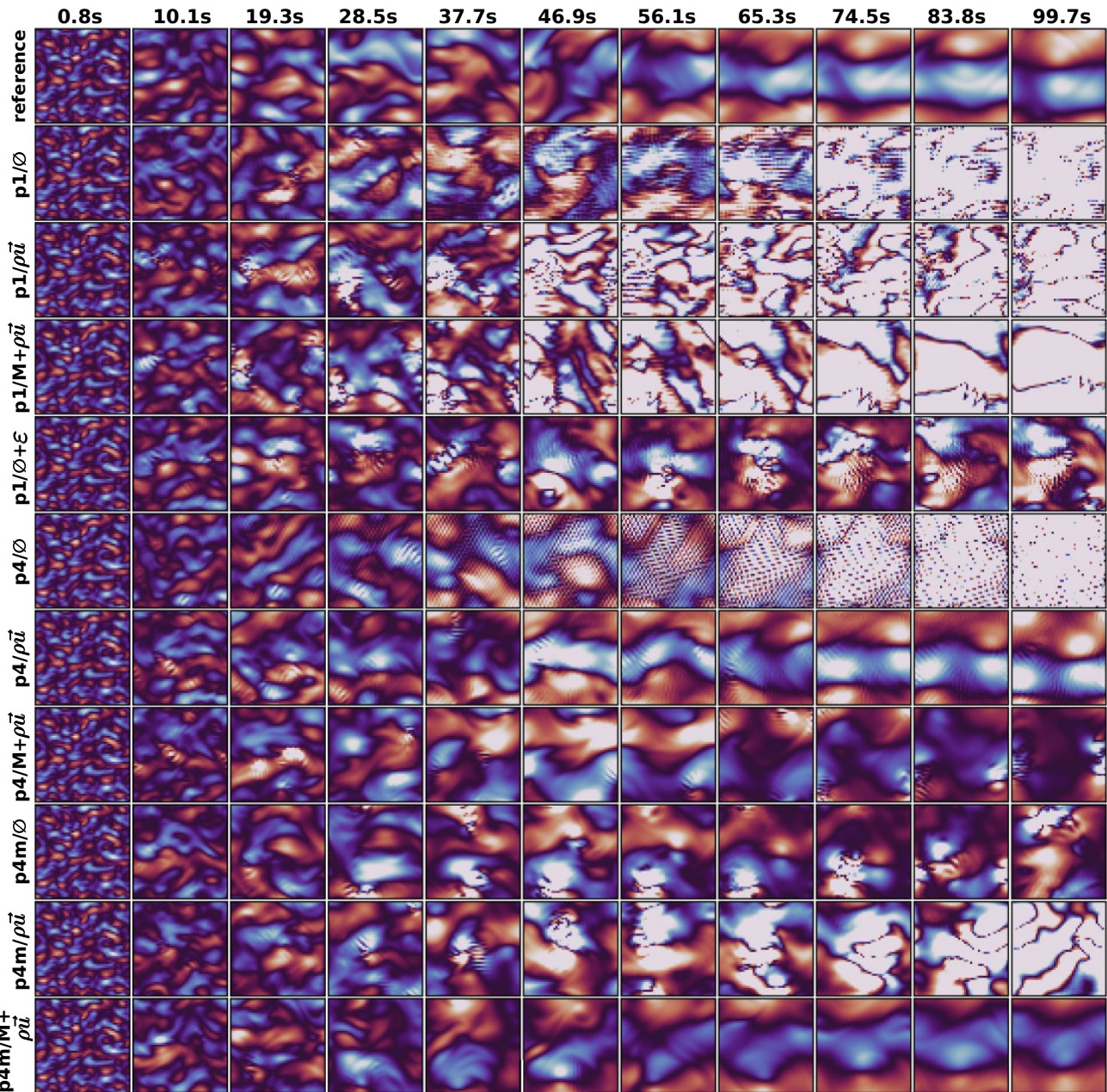

*Figure 32.* Rollout performance of networks with various physical and symmetry constraints for the generalization test of decaying turbulence. The figures depict the evolution of the field variable $u$.

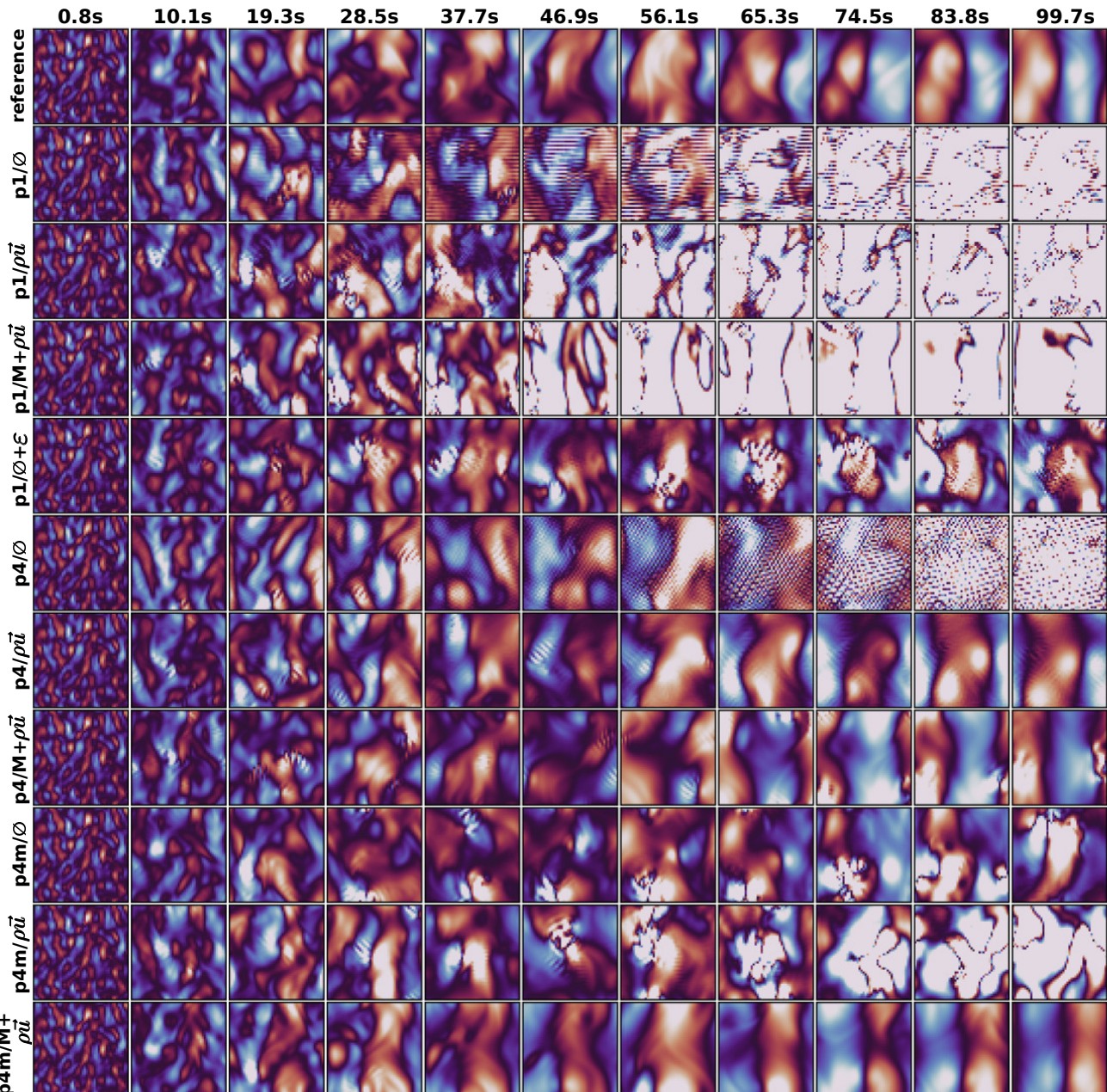

*Figure 33.* Rollout performance of networks with various physical and symmetry constraints for the generalization test of decaying turbulence. The figures depict the evolution of the field variable $v$.

Fig. 34 shows velocity and energy spectra for the reference solution and surrogates in the generalization test for INS. These spectra are expanded versions of those in Fig. 5e-f in the main text.p4m/M+$\rho\vec{u}$ (blue curves) exhibits the most precise alignment with the reference spectra (black) for both velocity and energy.

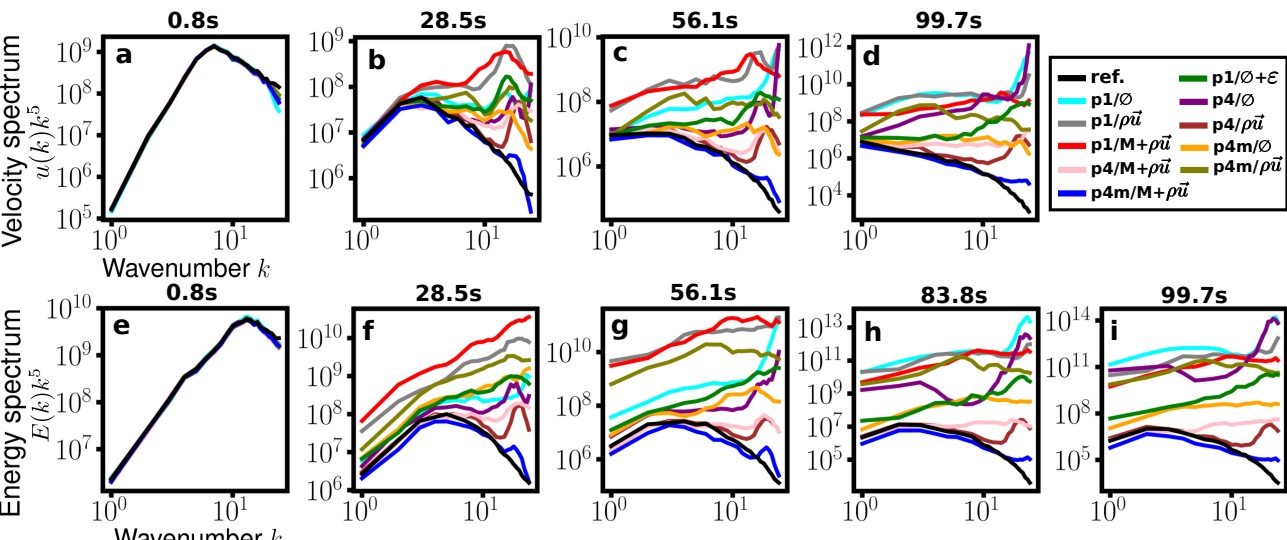

*Figure 34.* Velocity ($u$) and energy power spectra over an extended range of rollouts for the generalization of decaying turbulence case. This analysis extends the findings presented in Figure 5-(e,f) of the primary text. p4m/M+$\rho\vec{u}$ exhibits the best match to the energy and velocity spectra of reference solutions.

## J.12. Effects of network and dataset size on rollout performance

Fig. 35 show the effect of network and training dataset size on rollout performance, using INS surrogates based on modern U-net with 0.1M, 2M and 8.5M parameters. We consider training dataset sizes of 100, 400 and 760 unique ICs. The findings indicate that increasing the network size or the training dataset size enhances the rollout performance of the network. Additionally, networks with physical and symmetry constraints exhibits superior performance in each case.

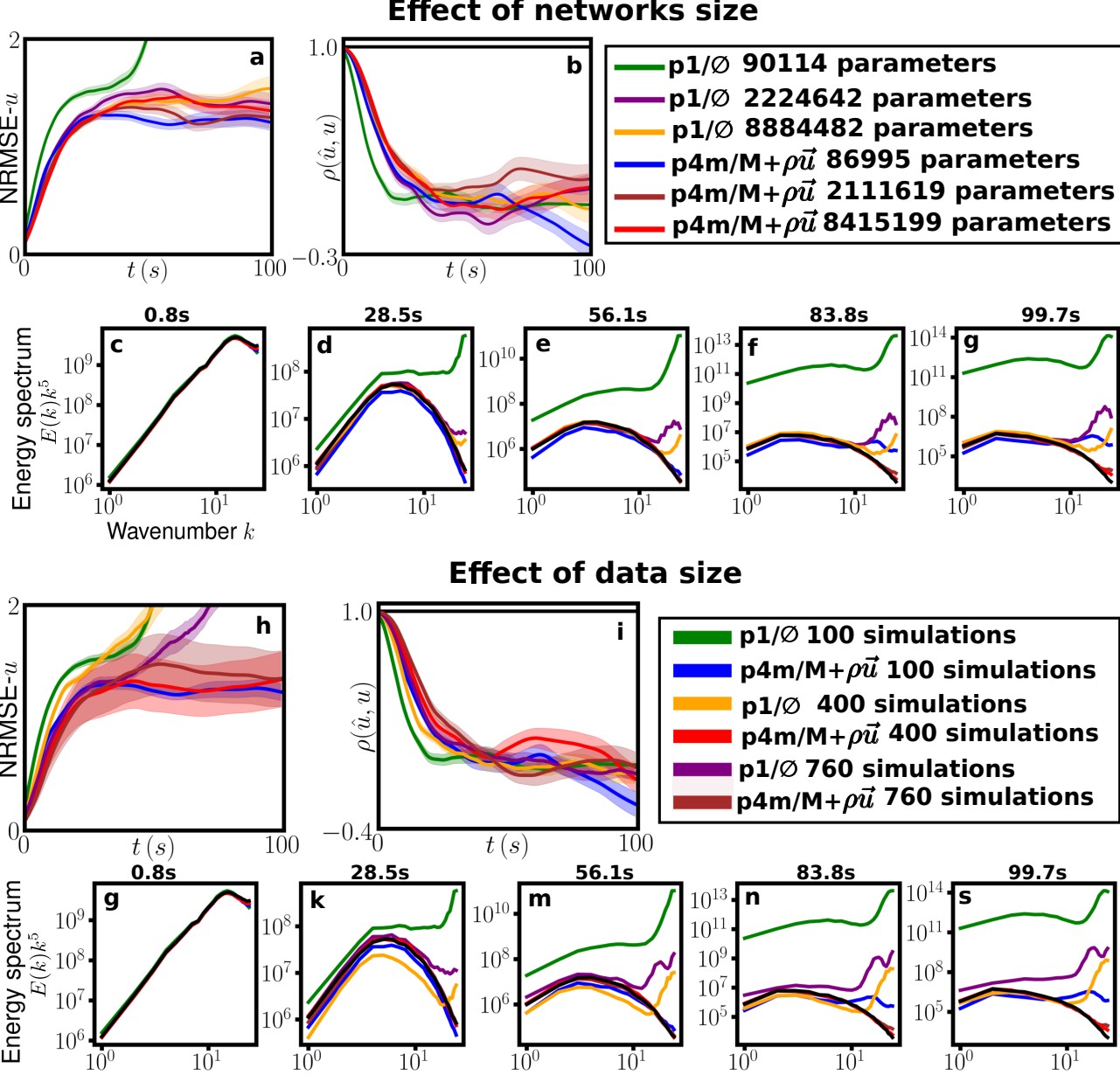

*Figure 35.* Top: effect of network size on NRMSE-$u$, $\rho(\hat{u}, u)$ (a, b) and the energy spectrum (c-g) for PDE surrogates. Bottom: effect of training data size on NRMSE-$u$, $\rho(\hat{u}, u)$ (h, i), and the energy spectrum (g,s). All results are reported for 99.7s rollouts.

### J.13. Inference time per time step of p1/∅ and p4m/M+$\rho\vec{u}$ at various network sizes

Table J.13 reports inference time per time step of p1/∅ and p4m/M+$\rho\vec{u}$ with various network sizes on CPU (Intel Xeon Platinum 8160) and GPU (Nvidia A100 40 GB) nodes. Inference speed was computed from an autoregressive rollout over 120 time steps. On the CPU, p4m/M+$\rho\vec{u}$ is consistently slower than the speed of p1/∅ for all network sizes, but on the GPU both networks are dominated by overhead costs that do not scale with network size up to 8.5 M parameters. Inference on the CPU is slower than on the GPU for both p1/∅ and p4m/M+$\rho\vec{u}$. These results indicate that the advantages conferred by physical and symmetry constraints do not adversely impact GPU-based predictions, at least for the domain and network sizes tested. The relative independence of GPU inference speed on network size suggests that overhead costs, kernel launches and memory transfers are likely bottlenecks, and further network scaling and optimization would be required to identify the true

costs of constrained vs. unconstrained inference in the limit of large network sizes.

*Table 9.* Inference time per time step of p1/$\varnothing$ and p4m/M+$\rho\vec{u}$ on CPU (Intel Xeon Platinum 8160) and GPU (nVidia A100 40 GB) nodes for various network sizes. Inference speed was computed for an autoregressive rollout over 120 time Steps.

| | p1/$\varnothing$ | | p4m/M+$\rho\vec{u}$ | |
| :---: | :---: | :---: | :---: | :---: |
| Networks size | CPU | GPU | CPU | GPU |
| 0.1 M | 9.4±0.9ms | 6.2±0.7ms | 20.6±3.2ms | 8.2 ±0.4ms |
| 2.0 M | 26.4±4.1ms | 6.3±0.3ms | 71.2±59.5ms | 12.0±3.3ms |
| 8.5 M | 27.5±3.8ms | 6.4±0.2ms | 145.7±98.7ms | 8.4±0.1ms |

### J.14. Real Ocean Dynamics

We used the ocean current velocity data was sourced from the Global Ocean Physics Analysis and Forecast (Marullo et al., 2014), and followed (Wang et al., 2021) for data selection and processing. The 6-hourly data from 2022-06-01 to 2025-04-05 was selected for analysis from three different ocean regions. The corresponding latitude and longitude ranges for these regions are listed below: (30 -42, 168 -180), (-49 -37, 80 -92), and (-51 -39, -30 -18). The data set under consideration has $144 \times 144$ pixel velocity fields. In order to reduce the size of the training set, it was downscaled to a coarse grid of $48 \times 48$ for training, testing and validation purposes.

Once surrogate training was completed, forecasts were made from a different region and time period. The latitude and longitude ranges for prediction were (-44 -32, -130 -118), and the time range was 2023-06-01 to 2025-04-05.

