# OpenReview forum: "Geometric and Physical Constraints Synergistically Enhance Neural PDE Surrogates"
_ICML.cc/2025/Conference — ICML 2025 poster_

### Official Review · Reviewer_hyP8 · 2025-03-03

**Overall Recommendation:** 5

**Summary:**

The authors propose a neural PDE surrogate solver that respects the rotation and reflection equivariance (via p4/p4m symmetry groups) and enforces physical conservation principles. Their approach is designed for scalar and vector field magnitudes on staggered grids, leveraging a modern U-Net architecture with group convolutions [Cohen & Welling, 2016]. Equivariance is enforced through custom input and output layers, while mass and momentum conservation are achieved by predicting a vector potential (for divergence-free conditions) and applying global mean corrections. The method is evaluated on two PDE systems: shallow water equations and incompressible decaying turbulence. Compared to baselines without these constraints or relying on data augmentation, the proposed model demonstrates improved accuracy and long-term stability.

===========

Post-rebuttal: I think the authors have addressed all my issues, and the work represents a neat investigation of equivariance for U-net architectures. There are really novel aspects here, and the approach seems useful. Hence, I fully support an accept of this paper for ICML.

**Claims And Evidence:**

The authors claim that incorporating symmetry constraints (rotation/reflection equivariance) and physical conservation laws (mass/momentum conservation) improves the accuracy and stability of neural PDE surrogates. They further argue that combining both constraints leads to the best performance, even compared to strong baselines that use data augmentation or pushforward training.

Their empirical results support these claims, with particularly notable improvements in long-term stability.

However, while the results are compelling, comparisons to other equivariant models (e.g., [Wang et al., 2020]) are missing. Additionally, this method is likely slower than a non-equivariant U-Net with the same number of weights. To provide a fairer assessment of practical trade-offs, the authors could compare their model against a non-equivariant counterpart with an equivalent inference time instead.

**Essential References Not Discussed:**

This work primarily focuses on p4m equivariance on grids, but a large body of research exists on E(2)/E(3) and SE(2)/SE(3) equivariance in unstructured grids. The following references are relevant but not discussed:
	•	General equivariance methods:
[1] Thomas et al. (2018) – Tensor Field Networks for rotation and translation-equivariant neural networks on 3D point clouds.
[2] Gasteiger et al. (2020) – Directional message passing for molecular graphs.
[3] Brandstetter et al. (2021) – Improving E(3)-equivariant message passing with geometric and physical constraints.
	•	Fluid dynamics applications:
[4] Lino et al. (2022) – Multi-scale rotation-equivariant GNNs for unsteady Eulerian fluid dynamics.
[5] Toshev et al. (2023) – E(3)-equivariant GNNs for particle-based fluid mechanics.

**Experimental Designs Or Analyses:**

The selected test problems demonstrate the method’s ability to enforce symmetry and conservation laws, but more complex problems could have been chosen to illustrate the practical relevance of p4m symmetries. For example, testing on fluid systems with irregular geometries or obstacles would better highlight the benefits of rotation equivariance.

**Methods And Evaluation Criteria:**

The method for guaranteeing p4m symmetry is based on an older approach [Cohen & Welling, 2016]. More recent methods ensure equivariance to continuous rotations (e.g., Tensor Field Networks), rather than just discrete ones, which may be more suitable for physics-based applications. Additionally, the advantage of using a staggered grid over the more common approach with CNNs is not clearly justified.

It should also be more explicitly stated that momentum conservation is enforced in an integral sense over the whole fluid domain, rather than pointwise. Achieving pointwise conservation might lead to even better results.

The experiments chosen to validate the method cover different physics and boundary conditions and are well explained. However, the fluid domains considered are geometrically simple. Since rotation equivariance is a key aspect of the method, it would be more relevant to test shallow water equations with walls or obstacles of varying geometries (e.g., [Simulating Surface Wave Dynamics with Convolutional Networks, Lino et al., 2020]). This would better assess how well the method generalizes to more complex real-world scenarios.

**Other Comments Or Suggestions:**

None.

**Other Strengths And Weaknesses:**

Strengths:
	•	The combination of equivariance and conservation laws leads to significant accuracy improvements.
	•	The results are strong, demonstrating better stability than many standard neural PDE solvers.
	•	The supplementary material is detailed.


Weaknesses:
	•	Lack of comparisons to other equivariant models.
	•	Computational efficiency is not analyzed—equivariant models are usually more expensive, and comparisons at equal inference time would provide a better practical assessment.
	•	Simple geometric test cases. Applying this method to more complex fluid domains (e.g., irregular obstacles) would better showcase its advantages.

**Questions For Authors:**

1.	How does the use of staggered grids improve performance compared to standard CNN approaches?
2.	Could momentum conservation be enforced pointwise instead of in an integral sense? Would that improve accuracy?
3.	What is the computational cost of your method compared to a standard CNN with an equivalent number of parameters?

**Relation To Broader Scientific Literature:**

The authors leverage discrete group convolutions in the hidden layers [Cohen & Welling, 2016] and modify the input layer to properly handle vector magnitudes. However, this is not a major innovation.

In fluid dynamics, Wang et al. (2020) have already incorporated symmetries and conservation principles using CNNs, and a broader body of literature has investigated rotation equivariance with Graph Neural Networks (GNNs). While the experimental results are strong, a key limitation is the lack of comparison with these prior methods, which would provide a clearer assessment of their contributions.

**Theoretical Claims:**

The design of the input layers is well justified and clearly explained in Appendix C. Overall, I find the theoretical claims to be well-founded, with the exception of two statements that seem unclear:
- “Boundary effects interfere with translation equivariance, so we provide a boundary mask input channel.”
- “Momentum is not conserved due to reflection from closed boundaries.”
It would be helpful if the authors could clarify these points.

---

> ### Author Rebuttal · Authors · 2025-03-31
>
> We appreciate the careful reading, positive assessments and constructive feedback.
>
> > comparisons to other equivariant models (e.g., [Wang et al., 2020]) are missing.
>
> We now compare to the equivariant network of Wang et al., 2020 on our simulation-based INS task, [updating fig. 4](https://tinyurl.com/2j8txw3n). We also compare to it on the real-world ocean current dataset used in that paper, with a [new figure](https://tinyurl.com/3u4a8wnu) and [table](https://tinyurl.com/mvz9u7f6).
>
> > Additionally, this method is likely slower than a non-equivariant U-Net with the same number of weights. To provide a fairer assessment of practical trade-offs, the authors could compare their model against a non-equivariant counterpart with an equivalent inference time instead.
>
> We now include inference speed of unconstrained and doubly-constrained networks in a new [table](https://tinyurl.com/44w2aubv). CPU implementations of equivariant networks were about half as fast, and inference time grew sublinearly with network size. For GPU implementations, no consistent relationship was observed, and equivariant networks were about 30% slower. These results suggest that overhead costs, kernel launches and memory transfers are likely bottlenecks, and further analysis and optimization would be required to fairly and quantitatively compare true inference speeds. We now discuss these issues further in our discussion section.
>
> > Unclear: “Boundary effects interfere with translation equivariance, so we provide a boundary mask input channel.”
>
> We agree and have moved this sentence to the previous paragraph, and revised to: "Since the time evolution of this SWE systems depends on the location of boundaries, we provide a binary boundary mask to the network as an additional input field with scalar values defined at grid cell centers. We note that this binary mask is invariant to rotations and reflections."
>
> > Unclear: “Momentum is not conserved due to reflection from closed boundaries.” It would be helpful if the authors could clarify these points.
>
> Revised for clarity: "Momentum is not conserved in this SWE system, and a wave travelling eastward will reverse and head westwards after reflecting from a boundary. In reality, this momentum change would be compensated by a slight change in the momentum of the Earth itself, but this is not modeled in our simulation."
>
> > The advantage of using a staggered grid over the more common approach with CNNs is not clearly justified.
>
> Staggered grids are prevalent in atmospheric [1] and ocean models [2], largely due to the numerical advantages of finite volume approaches, especially for conservation laws [3]. We now discuss these advantages in greater detail in the introduction.
>
> > This work primarily focuses on p4m equivariance on grids, but a large body of research exists on E(2)/E(3) and SE(2)/SE(3) equivariance in unstructured grids. The following references are relevant but not discussed: ...
>
> We agree and now included them all, along with several papers from 2024-2025.
>
> > Simple geometric test cases. Applying this method to more complex fluid domains (e.g., irregular obstacles) would better showcase its advantages.
>
> We agree that more complex geometries, as well as the challenge of generalization to new geometries (c.f. Wandel et al., 2020), are important open questions. While beyond our current scope, we now discuss this explicitly in "limitations and future work."
>
> > How does the use of staggered grids improve performance compared to standard CNN approaches?
>
> We do not claim staggered grids are always better, and our networks do not use them for the internal representations of hidden layers. But when input data/target outputs are defined on a staggered grid, only input/output layers taking this into account can maintain equivariance, and standard libraries such as escnn will not do so. Thus, to the extent that equivariance improves performance, staggered grids must be accounted for.
>
> > Could momentum conservation be enforced pointwise instead of in an integral sense? Would that improve accuracy?
>
> We suspect this would improve performance, especially when generalizing to new domain sizes. It would also ensure that surrogates learn causal relationship instead of memorizing statistical patterns. We expanded our discussion of this in appendix E, and now mention it in the discussion.
>
> > What is the computational cost of your method compared to a standard CNN with an equivalent number of parameters?
>
> We address this in a [new table](https://tinyurl.com/44w2aubv) (see answer above).
>
> [1] Giorgetta, et al. "ICON‐A, the atmosphere component of the ICON earth system model: I. Model description." Journal of Advances in Modeling Earth Systems, 2018.
>
> [2] NEMO Ocean Engine Reference Manual v 5.0. Madec et al., 2024.
>
> [3] Ferziger, Joel H., Milovan Perić, and Robert L. Street. Computational methods for fluid dynamics. springer, 2019.

---

### Official Review · Reviewer_tGcM · 2025-03-14

**Overall Recommendation:** 3

**Summary:**

The paper explores how incorporating symmetric constraints and physical priors can improve predictions within the same base architecture. Specifically, it investigates the effects of integrating additional symmetry equivariance into convolutions—such as rotation and reflection—in combination with conservation laws (e.g., mass and momentum conservation) that align with the underlying dynamics. The study employs also complementary techniques like data augmentation and the push-forward trick, where backpropagation is restricted to the final step, to enhance performance.

**Claims And Evidence:**

- Comprehensive study on symmetries and prior knowledge: The paper systematically examines the extent to which different combinations of symmetries and physical priors influence prediction performance, both within the training trajectory horizon and in extrapolation beyond it.
- Enhanced generalization: By integrating symmetry constraints and physical priors, the approach improves the model’s ability to generalize beyond the training data.

**Essential References Not Discussed:**

The related work is sufficient comprehensive.

**Experimental Designs Or Analyses:**

Yes.

**Methods And Evaluation Criteria:**

- Validity of the methods: The proposed methods are reasonable when the physical prior is known, ensuring that the imposed constraints align with the underlying dynamics.
- Empirical evaluation: The study employs two synthetic datasets to systematically analyze the effects of different symmetry and prior combinations on prediction performance.

**Other Comments Or Suggestions:**

- The study could benefit from validation on real-world datasets, such as sea surface temperature or atmospheric dynamics, to assess the generalizability of the proposed methods.

**Other Strengths And Weaknesses:**

Strengths
- Compared to previous works, particularly Wang et al. (2020), this study conducts a more extensive analysis with finer-grained comparisons of different symmetry and prior combinations.

Weaknesses
- Unlike Wang et al. (2020), the study does not include real-world data, limiting its direct applicability to practical scenarios.


References:
- Wang et al. (2020), Incorporating symmetry into deep dynamics models for improved generalization.

**Questions For Authors:**

See suggestions above.

**Relation To Broader Scientific Literature:**

The key contributions of the paper is related to the integration of prior information for neural surrogate model learning.

**Theoretical Claims:**

No.

---

> ### Author Rebuttal · Authors · 2025-03-31
>
> We thank the reviewer for the careful evaluation and appreciate the positive assessments therein.
>
> We have revised the manuscript to incorporate the real-world dataset from Wang et al. 2020, and added a new [figure](https://tinyurl.com/3u4a8wnu) and [table](https://tinyurl.com/mvz9u7f6). Similar to our results on simulation-based datasets, we find equivariant and physically constraints networks are more accurate than the same archictures without constraints and similar parameter counts. We achieved better accuracy than the equivariant network proposed in Wang et al. 2020, and also outperform it on our Naiver Stokes task, [updating fig. 4](https://tinyurl.com/2j8txw3n).

---

### Official Review · Reviewer_YKe6 · 2025-03-14

**Overall Recommendation:** 3

**Summary:**

The authors propose new input layers that can add inductive symmetry and conservation-law biases to neural PDE solvers to improve their performance in long-term rollouts. The main innovation of the work seems to be the ability to accommodate staggered grids commonly found in CFD. Other than this, the novelty component of the work is low. Its main contribution is a high-quality scientific computation study of tough CFD problems (the shallow water equation with close boundaries and decaying incompressible turbulence) using neural PDE solvers.

**Claims And Evidence:**

Yes, the claims are supported by high-quality experiments.

**Essential References Not Discussed:**

None noted.

**Experimental Designs Or Analyses:**

I checked briefly the experimental design and it seems adequate.

**Methods And Evaluation Criteria:**

The authors evaluate their approach on two difficult CFD problems.

**Other Comments Or Suggestions:**

Nothing to add here.

**Other Strengths And Weaknesses:**

The paper is clearly written in a adequate technical style.

**Questions For Authors:**

Would the use of FORTRAN be consistent with this kind of computationally-intensive data-driven work? I am just curious.

Can the authors define more clearly what they mean by "equivariance"? Would that be symmetry-group invariance?

**Relation To Broader Scientific Literature:**

The authors do a good job positioning the work with respect to the literature.

**Theoretical Claims:**

There aren't any theoretical claims.

---

> ### Author Rebuttal · Authors · 2025-03-31
>
> We thank the reviewer for the positive assessment of our work and the recognition that we used challenging tasks.
>
> We disagree, however, that the novelty component of the work is low overall. It is certainly true that the methods we introduce, equivariant input and output layers for staggered grids, are not revolutionary. However, we would argue that the major novelty and signficance of our work lies in our results themselves: it was known that both physical and symmetry constraints can improve long term accuracy of PDE surrogates, but to date there have been practically no results showing how well these approaches could be combined. Indeed, at the start of this project we were fully prepared for a negative result, in which one set of constraints made the other redundant. This would be, intuitively at least, consistent with the deep connection between symmetries and conservation laws in physics expressed by Noether's theorem (which admittedly does not hold on for discrete symmetry groups), and the fact that most numerical solvers are equivariant. In the end, we arrived at several important discoveries:
> * We can indeed fruitfully combine both constraint types.
> * We can go even further by combining them with other strategies for long-term accuracy such as pushforward training (Fig. 4g).
> * These results hold for multiple network sizes and architectures.
> * These results extend to real-world observational data.
> * The combination of symmetry and physical constraints improves generalization (Fig. 5).
>
> These results are promising and of high relevance for PDE surrogate tasks requiring long-term accuracy or where training data is in short supply, such weather forecasting, climate projections and airfoil design. We now better emphasize the importance of these novel results in the introduction and discussion.
>
> > Would the use of FORTRAN be consistent with this kind of computationally-intensive data-driven work? I am just curious.
>
> The constraints we have described are applicable in any programming language. ML in FORTRAN has several modern libraries and bindings, and can exchange data with python processes [1,2,3].
>
> > Can the authors define more clearly what they mean by "equivariance"? Would that be symmetry-group invariance?
>
> We say a function is equivariant when it respects a set of symmetry constraints. That is we, refer to equivariance with respect to a group of symmetries specific to a PDE, as described in the "symmetry equivariance" paragraph of section 2 (see eq. 3, with specific examples in eq. 4-5). Following Cohen & Welling [4], we define equivariance for a PDE surrogate $\mathcal M:w^t\rightarrow w^{t+1}$ as the relation $\mathcal T_g \circ \mathcal M(w^t) = \mathcal M (\mathcal T_g w^t), \forall g\in G$. Here $G$ is a group of symmetry transforamtions, and $\mathcal T_g$ is the transformation on the set of PDE fields described by the group element $g$ (for example, a ninety-degree rotation). Equivariance means that every transformation of the inputs of $\mathcal M$ leads to a corresponding transformation of the outputs. Equivalently, we say that $\mathcal M$ respects the symmetries described by $G$. In general we try to endow our surrogates with the same symmetries as the numerical solvers and PDEs they are learning from.
>
> We have added the clarifying sentence to the "symmetry equivariance" paragraph of sec. 2: "That is, transforming the inputs of $f$ will transform its outputs correspondingly."
>
> [1] Brenowitz, Noah. Calling Python from Fortran (not the other way around). https://www.noahbrenowitz.com/post/calling-fortran-from-python/. 2022
>
> [2] Zhang, Tao, et al. "A Fortran-Python Interface for Integrating Machine Learning Parameterization into Earth System Models." Geoscientific Model Development Discussions 2024 (2024): 1-26.
>
> [3] Arnold, Caroline, et al. "Efficient and stable coupling of the SuperdropNet deep-learning-based cloud microphysics (v0. 1.0) with the ICON climate and weather model (v2. 6.5)." Geoscientific Model Development 17.9 (2024): 4017-4029.
>
> [4] Cohen T, Welling M. Group equivariant convolutional networks. International conference on machine learning 2016 Jun 11 (pp. 2990-2999). PMLR.

---

### Official Review · Reviewer_ehyY · 2025-03-18

**Overall Recommendation:** 3

**Summary:**

This paper propose to integrate rotation symmetry of staggered grid into PDE surrogate models. Additionally, the models also encode physics constraints in the network readout. The experiments are conducted on closed shallow water equations and decaying turbulence.

**Claims And Evidence:**

- The motivation of using staggered C-grid is clear and designing equivariant models for it is well-motivated.

- However, I feel the presentation of the method is not very clear. Since I am not very familiar with the staggered C-grid. The technical challenges of extending group CNN to staggered C-grid are not sufficiently highlighted for me to get a clear understanding. Moreover, how to enforce the conservation laws is only briefly mentioned in the method section.

**Essential References Not Discussed:**

- Not I am aware of.

**Experimental Designs Or Analyses:**

- Section 4.1: "Simulations in Fortran required 67 seconds on the CPU", is the time for simulating one trajectory or the training set?

- Table 3: from the result (especially at 25h) it looks like the baseline models barely work. Although I think symmetry is important but the effect is rather surprising.

**Methods And Evaluation Criteria:**

- The benchmarks make sense to me.

**Other Comments Or Suggestions:**

- The authors could consider moving figure 7b and description of staggered C-grid to the main text.

- Minor: is this the right paper template? The footnote at first page and the running title seem to be missing.

**Other Strengths And Weaknesses:**

- It is interesting to use closed boundary for SWEs, which make them more challenging.

- The baseline models are reasonably chosen.

**Questions For Authors:**

- Assuming one has a good understanding of group equivariant CNNs, how would the authors explain the extension to staggered C-grid?

- Are the physics conservation laws hard to implement? I wonder since it is not described too much in the method section but it is emphasized a lot in the experiments.

**Relation To Broader Scientific Literature:**

- They seem to be adequately discussed.

**Theoretical Claims:**

- They look good to me.

---

> ### Author Rebuttal · Authors · 2025-03-31
>
> We appreciate the careful reading and constructive feedback.
>
> > technical challenges of extending group CNN to staggered C-grid not sufficiently highlighted
>
> We agree and have revised and expanded the last sentence of the paragraph labeled "Staggered Grids" in sec. 2 to read as follows:
>
> "However, current software implementations of equivariant network layers cannot be applyed to PDE variable fields on equivariant grids. This is because they assume that variables fields are all located at the same points, allowing the action of symmetries on these fields to be broken into two steps: a resampling step $x \rightarrow g^{-1}x$ carried out on the grid itself, and a transformation step $w\rightarrow \rho_g(w)$  carried out on PDE field variables $w\in\mathbb R^m$ at each point single grid point. This leads to overall transformations $w\rightarrow \mathcal T_g w$, such that $[\mathcal T_g w] (x) = \rho_g(w(g^{-1}x))$. This is a valid assumption for PDEs on continous spatial domains (eq. 5) or for colocated grids (Weiler, 2021, eq. 1). But for staggered grids, the PDE fields are not represented as a vector of values $w(x)$ at each grid point $x$. Instead, each field is defined at different locations, which may be grid cell centers, interfaces or verticies. Thus, the spatial transformation of the grid and the transformation of local field values cannot be disentangled. Applying existing equivariant network layers to staggered PDE fields therefore breaks symmetry constraints."
>
> We also added a [new supplementary figure](https://tinyurl.com/2rn8ydp5), showing how equivariance breaks down when applying previous equivariant input layers to input data on a staggered grid.
>
> > Moreover, how to enforce the conservation laws is only briefly mentioned in the method section.
>
> While space constraints prevent us from giving full details in the main text, we have revised the relevant material for clarity and to more explicitly list the types of constraints we can impose while maintianing equivariance on staggered grids. We now include in section 3 a summary of the full stragegy described in appendix E. The paragraph beginning "conservation laws" now reads:
>
> "We impose 3 types of conservation laws as hard constraints. For scalar quantities such as fluid surface height $\zeta$, we subtract the global mean of $\zeta^{t+1}-\zeta^t$ at each time step. For vector fields, we subtract the mean of each velocity component. As mass conservation in incompressible flows is equivalent to divergence-free velocity fields, we impose this by learning a vector potential $a$ defined at grid vertices, and compute velocities at grid cell interfaces as the curl $\nabla \times a$ to satisfy both mass and momentum conservation (Wandel et al., 2020). Further details and discussion of alternative approaches are found in appendix E."
>
> We also discuss other possible physical constraints in the discussion.
>
> > Section 4.1: "Simulations in Fortran required 67 seconds on the CPU", is the time for simulating one trajectory or the training set?
>
> For a one trajectory, now clarified.
>
> > Table 3: from the result (especially at 25h) it looks like the baseline models barely work. Although I think symmetry is important but the effect is rather surprising.
>
> This is a challenging task and no models tested were accurate over the full 50 simualted hours. NaN values in the table indicate that some examples from the test set diverged to infinity, which is now clarified. Symmetry was essential to achieving accurate results at 25h, but physical constraints were also important, as rotation-reflection equivariant without physical constraints also diverged within 25h. As shown in Fig. 3f, this can partly be explained by the fact that in many non-mass-conserving networks total mass diverged to infinity. We now mention this in the discussion.
>
> > The authors could consider moving figure 7b and description of staggered C-grid to the main text.
>
> We agree and plan to add additional detail on the staggered C-grid to fig. 1.
>
> > Minor: is this the right paper template? The footnote at first page and the running title seem to be missing.
>
> We appreciate the notice and will correct this if accepted.
>
> > Assuming one has a good understanding of group equivariant CNNs, how would the authors explain the extension to staggered C-grid?
>
> Staggered grids invalidate the assumption that symmetries can be brokedn into a transformation of space and a transformation of PDE fields at each point. Equivariance therefore requires new input/output layers that take staggering into account.
>
> > Are the physics conservation laws hard to implement? I wonder since it is not described too much in the method section but it is emphasized a lot in the experiments.
>
> We have added further detail to the methods. Conservation laws are easy to implement by correcting or re-interpreting network outputs, without considering network architecture or equivariance. We will provide code if accepted.

---

### Decision · Program_Chairs · 2025-05-01

**Decision:**

Accept (poster)

**Comment:**

This paper proposes novel input and output layers for neural PDE surrogates that enable the enforcement of physical constraints (conservation laws) and geometric constraints (symmetry equivariance) specifically on staggered grids, which are common in CFD. The authors systematically investigate the individual and combined effects of these constraints on two challenging problems: shallow water equations with closed boundaries and decaying incompressible turbulence. They demonstrate that combining both types of constraints improves accuracy, long-term stability, and generalization compared to strong baselines.

The authors' thorough rebuttal effectively addressed the main weaknesses identified by the reviewers. While some limitations remain the core findings on the power of combining constraints on staggered grids represent a solid contribution.